# Lives saved with vaccination for 10 pathogens across 112 countries in a pre-COVID-19 world

Jaspreet Toor[1†], Susy Echeverria-Londono[1†], Xiang Li[1†], Kaja Abbas[2], Emily D Carter[3], Hannah E Clapham[4], Andrew Clark[2], Margaret J de Villiers[1], Kirsten Eilertson[5], Matthew Ferrari[6], Ivane Gamkrelidze[7], Timothy B Hallett[1], Wes R Hinsley[1], Daniel Hogan[8], John H Huber[9], Michael L Jackson[10], Kevin Jean[1,11], Mark Jit[2,12], Andromachi Karachaliou[13], Petra Klepac[2], Alicia Kraay[14], Justin Lessler[3], Xi Li[15], Benjamin A Lopman[14], Tewodaj Mengistu[8], C Jessica E Metcalf[16], Sean M Moore[9], Shevanthi Nayagam[1,17], Timos Papadopoulos[18,19], T Alex Perkins[9], Allison Portnoy[20], Homie Razavi[7], Devin Razavi-Shearer[7], Stephen Resch[20], Colin Sanderson[2], Steven Sweet[20], Yvonne Tam[3], Hira Tanvir[2], Quan Tran Minh[9], Caroline L Trotter[13], Shaun A Truelove[3], Emilia Vynnycky[18], Neff Walker[3], Amy Winter[3], Kim Woodruff[1], Neil M Ferguson[1], Katy AM Gaythorpe[1]*

[1]MRC Centre for Global Infectious Disease Analysis; and the Abdul Latif Jameel Institute for Disease and Emergency Analytics (J-IDEA), School of Public Health, Imperial College London, London, United Kingdom; [2]London School of Hygiene and Tropical Medicine, London, United Kingdom; [3]Bloomberg School of Public Health, Johns Hopkins University, Baltimore, United States; [4]Saw Swee Hock School of Public Health, National University of Singapore, Singapore; Oxford University Clinical Research Unit, Vietnam; Nuffield Department of Medicine, Oxford University, Oxford, United Kingdom; [5]Colorado State University, Fort Collins, United States; [6]Pennsylvania State University, State College, United States; [7]Center for Disease Analysis Foundation, Lafayette, United States; [8]Gavi, the Vaccine Alliance, Geneva, Switzerland; [9]Department of Biological Sciences, University of Notre Dame, Notre Dame, United States; [10]Kaiser Permanente Washington, Seattle, United States; [11]Laboratoire MESuRS and Unite PACRI, Institut Pasteur, Conservatoire National des Arts et Metiers, Paris, France; [12]University of Hong Kong, Hong Kong Special Administrative Region, Hong Kong, China; [13]University of Cambridge, Cambridge, United Kingdom; [14]Rollins School of Public Health, Emory University, Atlanta, United States; [15]Independent, Atlanta, United States; [16]Princeton University, Princeton NJ, United States; [17]Section of Hepatology and Gastroenterology, Department of Metabolism, Digestion and Reproduction, Imperial College London, London, United Kingdom; [18]Public Health England, London, United Kingdom; [19]University of Southampton, Southampton, United Kingdom; [20]Center for Health Decision Science, Harvard T H Chan School of Public Health, Harvard University, Cambridge, United States

*For correspondence:
k.gaythorpe@imperial.ac.uk

[†]These authors contributed equally to this work

## Abstract

**Background:** Vaccination is one of the most effective public health interventions. We investigate the impact of vaccination activities for *Haemophilus influenzae* type b, hepatitis B, human papillomavirus, Japanese encephalitis, measles, *Neisseria meningitidis* serogroup A, rotavirus,

rubella, *Streptococcus pneumoniae*, and yellow fever over the years 2000–2030 across 112 countries.

**Methods:** Twenty-one mathematical models estimated disease burden using standardised demographic and immunisation data. Impact was attributed to the year of vaccination through vaccine-activity-stratified impact ratios.

**Results:** We estimate 97 (95%CrI[80, 120]) million deaths would be averted due to vaccination activities over 2000–2030, with 50 (95%CrI[41, 62]) million deaths averted by activities between 2000 and 2019. For children under-5 born between 2000 and 2030, we estimate 52 (95%CrI[41, 69]) million more deaths would occur over their lifetimes without vaccination against these diseases.

**Conclusions:** This study represents the largest assessment of vaccine impact before COVID-19-related disruptions and provides motivation for sustaining and improving global vaccination coverage in the future.

**Funding:** VIMC is jointly funded by Gavi, the Vaccine Alliance, and the Bill and Melinda Gates Foundation (BMGF) (BMGF grant number: OPP1157270 / INV-009125). Funding from Gavi is channelled via VIMC to the Consortium's modelling groups (VIMC-funded institutions represented in this paper: Imperial College London, London School of Hygiene and Tropical Medicine, Oxford University Clinical Research Unit, Public Health England, Johns Hopkins University, The Pennsylvania State University, Center for Disease Analysis Foundation, Kaiser Permanente Washington, University of Cambridge, University of Notre Dame, Harvard University, Conservatoire National des Arts et Métiers, Emory University, National University of Singapore). Funding from BMGF was used for salaries of the Consortium secretariat (authors represented here: TBH, MJ, XL, SE-L, JT, KW, NMF, KAMG); and channelled via VIMC for travel and subsistence costs of all Consortium members (all authors). We also acknowledge funding from the UK Medical Research Council and Department for International Development, which supported aspects of VIMC's work (MRC grant number: MR/R015600/1).

JHH acknowledges funding from National Science Foundation Graduate Research Fellowship; Richard and Peggy Notebaert Premier Fellowship from the University of Notre Dame. BAL acknowledges funding from NIH/NIGMS (grant number R01 GM124280) and NIH/NIAID (grant number R01 AI112970). The Lives Saved Tool (LiST) receives funding support from the Bill and Melinda Gates Foundation.

This paper was compiled by all coauthors, including two coauthors from Gavi. Other funders had no role in study design, data collection, data analysis, data interpretation, or writing of the report. All authors had full access to all the data in the study and had final responsibility for the decision to submit for publication.

## Introduction

Vaccines play a vital role in immunising populations worldwide to provide protection against a wide range of diseases. In 1974, the World Health Organisation (WHO) launched the Expanded Programme on Immunisation (EPI) with a goal of universal access to all relevant vaccines for all at risk (*Keja et al., 1988*). To further increase momentum on vaccine coverage, Gavi, the Vaccine Alliance, was created in 2000 with a goal of providing vaccines to save lives and protect people's health (*Bill & Melinda Gates Foundation, 2020*; *Zerhouni, 2019*). Over the past two decades, vaccination programmes have expanded across low- and middle-income countries (LMICs), significantly reducing morbidity and mortality related to vaccine preventable diseases (VPDs). As of 2019, Gavi has helped immunise over 822 million children through routine programmes and provided over 1.1 billion vaccinations through campaigns in supported countries (*Gavi, the Vaccine Alliance, 2019*). Despite this immense progress, almost one in five (15.2 million) children in Gavi-supported countries remain under-immunised with the third dose of the essential childhood vaccination containing diphtheria-tetanus-pertussis vaccine (DTP3), 10.6 million of these children are zero-dose children, that is, having not received their first dose of DTP (*Zerhouni, 2019*).

The beneficial effect of vaccination programmes cannot be assessed directly as the counterfactual, that is, the situation without vaccination, cannot be observed. Hence, models of disease risk and the impact of vaccination activities play a vital role in assessing the current burden, examining the effect of previous activities and projecting the future situation. The Vaccine Impact Modelling

Consortium (VIMC), established in 2016, aims to deliver an effective, transparent and sustainable approach to generating disease burden and vaccine impact estimates (*Imperial College London, 2021*). The VIMC consists of twenty-one independent research groups which provide estimates of disease burden and vaccine impact across 112 LMICs for 10 pathogens, namely hepatitis B (HepB), *Haemophilus influenzae* type b (Hib), human papillomavirus (HPV), Japanese encephalitis (JE), measles, *Neisseria meningitidis* serogroup A (MenA), *Streptococcus pneumoniae* (PCV), rotavirus (Rota), rubella, and yellow fever (YF).

There are various ways of calculating the impact of vaccination (*Echeverria-Londono et al., 2021*). The burden averted by vaccination can be estimated in terms of the number of cases, deaths and disability adjusted life years (DALYs) averted. Vaccine impact is commonly presented by calendar year, that is, the number of lives saved by vaccination in a particular year or by birth cohort, that is, the number of lives saved by vaccination over the lifetime of individuals born in a particular year. Previous work by the VIMC on these 10 pathogens estimated that 69 million deaths would be averted by vaccination over calendar years 2000–2030 across 98 LMICs, with 120 million deaths averted over the lifetime of birth cohorts born between 2000 and 2030 (*Li et al., 2019*). The WHO estimates that immunisation currently prevents 2–3 million deaths every year (*World Health Organisation, 2021*), similarly *Ehreth, 2003* estimated 3 million deaths averted due to vaccination for pathogens such as measles, YF, HepB, diptheria, Hib, pertussis, neonatal tetanus and poliomyelitis.

Although attributing vaccine impact to calendar year or birth cohort is intuitive and commonly used, these methods fail to capture the impact of a specific year's vaccination activities traced over the lifetime of those vaccinated. It is beneficial to examine the impact corresponding to a vaccination activity so that the cost and benefit of each intervention can be appropriately calculated. The impact by year of vaccination activity method, developed by the VIMC, estimates the number of individuals that will be saved due to a particular year's vaccination activities (*Echeverria-Londono et al., 2021*). This method addresses the issue of attributing impact to the vaccination activity that caused it without repeatedly rerunning modelling scenarios which, whilst the optimal approach, is extremely computationally intensive. As such, we can approximate the potential effect of one year's worth of activity.

The first human case of coronavirus 2019 (COVID-19) was reported in December 2019 and has subsequently affected vaccination and healthcare worldwide. Whilst the effect of COVID-19 is not the focus of the current study, we acknowledge the huge influence the global pandemic has had and will have for years to come. Preliminary work has begun on quantifying the effect of disruption on vaccination activities and on assessing the benefit of continuing routine infant immunisation in times of COVID-19 (*Abbas et al., 2020a*; *Gaythorpe et al., 2021b*). There is also evidence that the rise in non-pharmaceutical interventions (NPIs, e.g. social distancing) associated with the pandemic may reduce the transmission of certain pathogens, such as those that cause bacterial meningitis (*Taha and Deghmane, 2020*). However, there is also a risk that NPIs may result in a build up of susceptible individuals in the population, particularly for outbreak prone diseases, such as measles, but catch-up activities may be able to prevent this. Currently, there is little data to inform how the pandemic may have influenced long-term population health and vaccine coverage. In order to assess this, we need to firmly understand the impact of vaccination before the pandemic; only then will it be possible to assess changes due to this global disruption.

In this paper, we estimate the impact of immunisation by year of vaccination for the 10 pathogens modelled by the VIMC across 112 LMICs over the years 2000–2030. Burden averted is investigated in terms of deaths and DALYs averted in synthetic coverage scenarios (with vaccination) compared to counterfactual coverage scenarios (with no vaccination). Although the current COVID-19 pandemic may have hindered vaccination activities, our analyses focus on the projections given what has happened in the past (2000–2019) and given no disruption (from 2020 onward) thus presenting vaccine impact estimates prior to COVID-19.

## Materials and methods

### Models

The VIMC consists of multiple modelling groups. These provide disease-specific vaccine impact projections to a central Secretariat based at Imperial College London who then synthesise these

estimates. Twenty-one mathematical models were used to inform the estimates with two models per pathogen (except HepB which has three models) thereby increasing robustness and capturing structural uncertainty within the analyses. There is substantial variation in modelling approach due to both the differences in pathogen dynamics and inherent uncertainties in modelling disease risk. The model characteristics vary in their type, from static cohort to transmission-dynamic models; their complexity, for example in their representation of age effects; and their calibration and validation methods. A brief overview of pathogens is provided in *Table 1* with detailed model descriptions provided in Appendix 2.2 (HepB [*Nayagam et al., 2016*], HPV [*Goldie et al., 2008*; *Abbas et al., 2020b*], Hib [*Clark et al., 2019a*; *Walker et al., 2013a*], JE [*Quan et al., 2020*], Measles [*Chen et al., 2012*], MenA [*Karachaliou et al., 2015*; *Tartof et al., 2013*], PCV [*Walker et al., 2013a*; *Clark et al., 2019a*], Rota [*Pitzer et al., 2012*; *Clark et al., 2019a*], Rubella [*Boulianne et al., 1995*; *Vynnycky et al., 2019*], YF [*Gaythorpe et al., 2021a*]).

Each modelling group provided estimates of age-stratified disease burden at national level for three scenarios: no vaccination, only routine vaccination (routine immunisations; RI) and, where appropriate, both RI and non-routine vaccination (non-routine immunisations; NRI, such as multi-age cohort vaccinations for HPV, and catch-up campaigns for measles). Disease burden was quantified in terms of deaths and DALYs. DALYs measure the years of healthy life lost due to premature death and disability from the disease, and are the sum of years of life lost (YLLs) through premature mortality and years lived with disability (YLDs). No discounting or weighting was applied in the calculation of DALYs. For rubella, only disease burden from congenital rubella syndrome (CRS) was included and the models differed in the inclusion of deaths due to stillbirths.

For every pathogen, the modelling teams were asked to provide 200 samples of their burden estimates for each year, vaccination scenario, and country constructed from the probabilistic ranges of their model parameters. The same randomly sampled sets of parameters were used for the no vaccination and with vaccination model runs allowing the direct comparison of the estimates. In order to calculate the mean and credible interval (CrI) for each pathogen, the full probabilistic distributions of impact are combined from all models for a pathogen, then the mean and 95% CrI are calculated from the full distribution. Similarly, when calculating the aggregated impact across pathogens, bootstrap sampling was used. In these bootstraps, a sample of interest was taken from the individual model; this was then averaged across models of the same pathogen and then summed across all pathogens; finally, the mean and 95% CrIs were calculated from 1000 bootstrap samples.

**Table 1.** Vaccine Impact Modelling Consortium (VIMC) pathogen-specific details.

RI denotes routine immunisations and NRI denotes non-routine immunisations. RI schedule details the number of doses given and the ages (in years, y) targeted. Vaccination over 2000 - 2030 shows whether vaccination has been occurring over the years 2000 to 2030; years are shown where the vaccines have been introduced in later years. Countries included shows the maximum number of VIMC countries that had coverage in specific year(s) (coverage information in supplementary spreadsheet and countries listed in Appendix 6.1).

| Pathogen | Countries included | Activity type | RI schedule | Vaccination over 2000 - 2030 |
|---|---|---|---|---|
| Hepatitis B (HepB) | 112 | RI | Birth dose + Infant 3 doses (<1y) | Yes |
| Human papillomavirus (HPV) | 112 | RI + NRI | Adolescent girls 2 doses (9-14 y) | 2014–2030 |
| Haemophilus influenzae type B (Hib) | 112 | RI | Infant 3 doses (<1y) | Yes |
| Japanese encephalitis (JE) | 17 | RI + NRI | Infant dose (<1y) | 2005–2030 |
| Measles | 112 | RI + NRI | 1st dose (<=1 y) + 2nd dose (<2 y) | Yes |
| *Neisseria meningitidis* serogroup A (MenA) | 26 | RI + NRI | Infant dose (< 1 y) | 2010–2030 |
| *Streptococcus pneumoniae* (PCV) | 112 | RI | Infant 3 doses (<1y) | 2009–2030 |
| Rotavirus (Rota) | 112 | RI | Infant 2 doses (<1y) | 2006–2030 |
| Rubella | 112 | RI + NRI | 1st dose (< 1 y) + 2nd dose (< 2 y) | Yes |
| Yellow fever (YF) | 36 | RI + NRI | Infant dose (< 1 y) | Yes |

## Data and vaccination scenarios

Standardised, national-level, age-stratified demographic data was provided to all modellers from the 2019 United Nations World Population Prospects (UNWPP) for years 2000 to 2100 (*World Population Prospects, 2019*). The 112 countries considered here include 73 currently and formerly Gavi supported countries and 39 other countries that are of interest due to high burden and/or potential vaccine introduction. These 112 countries represent 99% of the total mortality attributed to measles for children under-5 using the WHO child causes of death 2000–2017 estimate (*World Health Organization, 2020*) and 96% of the total deaths attributed to measles, HepB, Hib, MenA, PCV, and YF of all ages using the Institute for Health Metrics and Evaluation (IHME) Global Burden of Disease Study (GBD) 2017 estimates (*GHDx, 2019*). Therefore, there has been a greater focus on supporting vaccine introduction and implementation in these countries, mainly through Gavi. Pathogens endemic only in certain regions such as JE, MenA, and YF have estimates for 17, 26, and 36 countries, respectively (*Table 1*).

For the vaccination scenarios, standardised vaccine coverage data were provided at a national level where past RI coverage (1980–2018) was obtained from WHO/UNICEF Estimates of National Immunisation Coverage (WUENIC) as published in July 2019 (*WHO UNICEF coverage estimates, 2020*). Historical campaign coverage (2000–2018) was taken from Gavi's data repository, which included data from various sources, mainly Gavi and WHO. For HPV, JE, MenA, PCV and Rota, RI and NRI were introduced later, from 2005 onward (*Table 1*). Future coverage estimates, both RI and NRI, from 2019 to 2030 were taken from default scenario forecasts, developed with Gavi, for all 112 countries (countries listed in Appendix 6.1). Projection for future (2030–2100) RI is done by assuming a 1% annual increment up to a threshold of 90% (95% for the first dose of measles containing vaccine, MCV1) or historical highest. We assume no campaigns post-2030 to avoid predicting future campaign coverage beyond the default scenario forecasts, see supplementary material for further details on coverage assumptions. Estimates of numbers of vaccines received per child were calculated based on these coverage estimates and projections assuming independence between vaccines.

In the no vaccination (counterfactual) scenario, zero coverage is assumed for all years from 1980 to 2100 except for YF which has historical reactive campaigns for outbreaks.

## Impact by year of vaccination

We calculate deaths and DALYs averted by year of vaccination using impact ratios stratified by vaccine activity type (*Echeverria-Londono et al., 2021*). In this way, we attribute the deaths averted due to vaccination to the year in which the vaccination activity took place. We stratify the impact ratios by activity type in order to account for the different effects of RI compared to NRI which has been found to better capture model projections (*Echeverria-Londono et al., 2021*). Hence, this method assumes vaccine impact varies between RI and NRI but does not vary across birth cohorts. This method averages the effects of any temporal changes in disease incidence or population health over the time period modelled. We present results using 'fully vaccinated persons (FVPs)' which refer to the total number of doses provided by a vaccination activity. Where separate coverage figures are provided, one vaccine dose results in one FVP. However, for diseases such as HepB, coverage figures are based on the completed courses of multi-dose vaccinations. More specifically, we also show deaths and DALYs averted per 1000 FVPs. Notably, for some of the pathogens, the different models assume varying levels of dose dependency. For example, the measles dynaMICE model assumes that NRI doses are weakly dependent on RI doses whereas the measles Pennsylvania State University model assumes that NRI doses are independent from prior RI doses and that the second dose (MCV2) is only given to those who received the first dose (MCV1) (further model details in Appendix 2.2). When assuming NRI doses are distributed randomly and thus may re-vaccinate some individuals, the relative benefit of NRI compared to RI, which will always vaccinate a naive individual, is affected.

As we model disease-specific mortality under different vaccination scenarios, when aggregating estimates of deaths averted across all 10 pathogens per calendar year or birth cohort, double counting can arise whereby an individuals' death is accounted for more than once. Under the year of vaccination method, we do not adjust death estimates for double counting.

### Impact by birth year for children under five

To investigate the impact of vaccination in children under-5, we calculate deaths and DALYs averted by birth cohort. Here, we aggregate the impact over the first 5 years of life of birth cohorts born within the years of interest and then calculate the difference in the no vaccination and with vaccination scenarios. Furthermore, in *Appendix 5—figure 1* and *Appendix 5—figure 2*, we present vaccine impact by calendar year and by birth cohort in line with *Li et al., 2019* which shows the impact in a particular year or the total impact over an individuals' lifetime, respectively. These methods are directly calculated through comparison of the focal scenario with vaccination (both RI and NRI where appropriate) to the counterfactual scenario without vaccination.

Within the birth cohort method when investigating the impact of vaccination in children under-5, we account for double counting of deaths attributable to the 10 VIMC pathogens. This is done by clustering a population or birth cohort by vaccine coverage and evaluating the proportion of disease burden in those un-vaccinated and vaccinated, respectively; via which the total deaths across all 10 pathogens is re-estimated with double counting removed. A full description of the method is given in *Echeverria-Londono et al., 2021*; *Li et al., 2019*.

## Results

### Estimated burden

The modelling groups produced estimates of deaths attributable to the pathogens for years 2000–2100 for the given vaccination scenarios. In the focal scenario with vaccination, coverage has improved over time leading to more FVPs (*Figure 2* and *Appendix 5—figure 3*). We find that given these improvements in coverage over time, there is a general decline in the mean number of predicted deaths due to the 10 VIMC pathogens in each of the 112 countries. The decline in deaths averted due to vaccination varies by country, largely due to variations in vaccination coverage over time as well as variation in the epidemiology, treatment assumptions, health access, case fatality ratio (CFR), pathogen-specific mortality and demographic parameters (e.g. life expectancy) of some pathogens by country. Without vaccination, there is still some reduction in deaths over time in some countries due to these latter factors (*Figure 1*). Notably, the total burden caused by these diseases disproportionately lies within the WHO African region where the greatest decline in burden is predicted (*Figure 1* and *Appendix 5—figure 4*).

The ages at which the greatest mortality risks are faced varies across the pathogens with mortality related to Hib, measles, Rota, rubella, and PCV mostly focused in children under-5 (*Appendix 5—figure 5* shows a corresponding decline in deaths in the under-5s when vaccination occurs). Mortality attributable to HepB and HPV is focused in those over 40, and for YF, MenA and JE this is focused in those under 30 (due to natural immunity acquired with age in older adults).

### Impact by year of vaccination

Due to vaccination activities over the years 2000–2030 for all 10 VIMC pathogens, 97 (95%CrI[80, 120]) million future deaths and 5100 (95%CrI[4100, 6300]) million DALYs are estimated to be averted. Focusing on the years prior to the COVID-19 pandemic, i.e. 2000 to 2019, 50 (95%CrI[41, 62]) million deaths and 2700 (95%CrI[2200, 3500]) million DALYs are estimated to have been averted. The remaining numbers averted arise from the years 2020 to 2030, which may be affected by COVID-19 and other changes to future vaccine introductions and coverage as well as changes in access to health care (*Table 2*). Note: although the first human case of COVID-19 was reported in December 2019, any effects of this on vaccination activities in 2019 would be negligible.

The Global Vaccine Action Plan (GVAP) target for 2011–2020 is to avert between 24 and 26 million future deaths with vaccination for the 10 pathogens over 94 countries (*World Health Organization, 2013*). Over 2011–2019, we estimate that 23 (95%CrI [19, 27]) million deaths will be averted, with this increasing to 26 (95%CrI [21, 31]) million deaths averted over 2011–2020 (without COVID-19-related disruptions in 2020). Hence, the achievement of the GVAP target will depend on how the year 2020 is impacted by COVID-19.

The years in which vaccination activities occur, the types of activities carried out, the coverage and the number of FVPs achieved varies by pathogen. Measles and HepB have activities occurring over the entire time period of interest from 2000 to 2030 and achieve higher coverage and FVPs

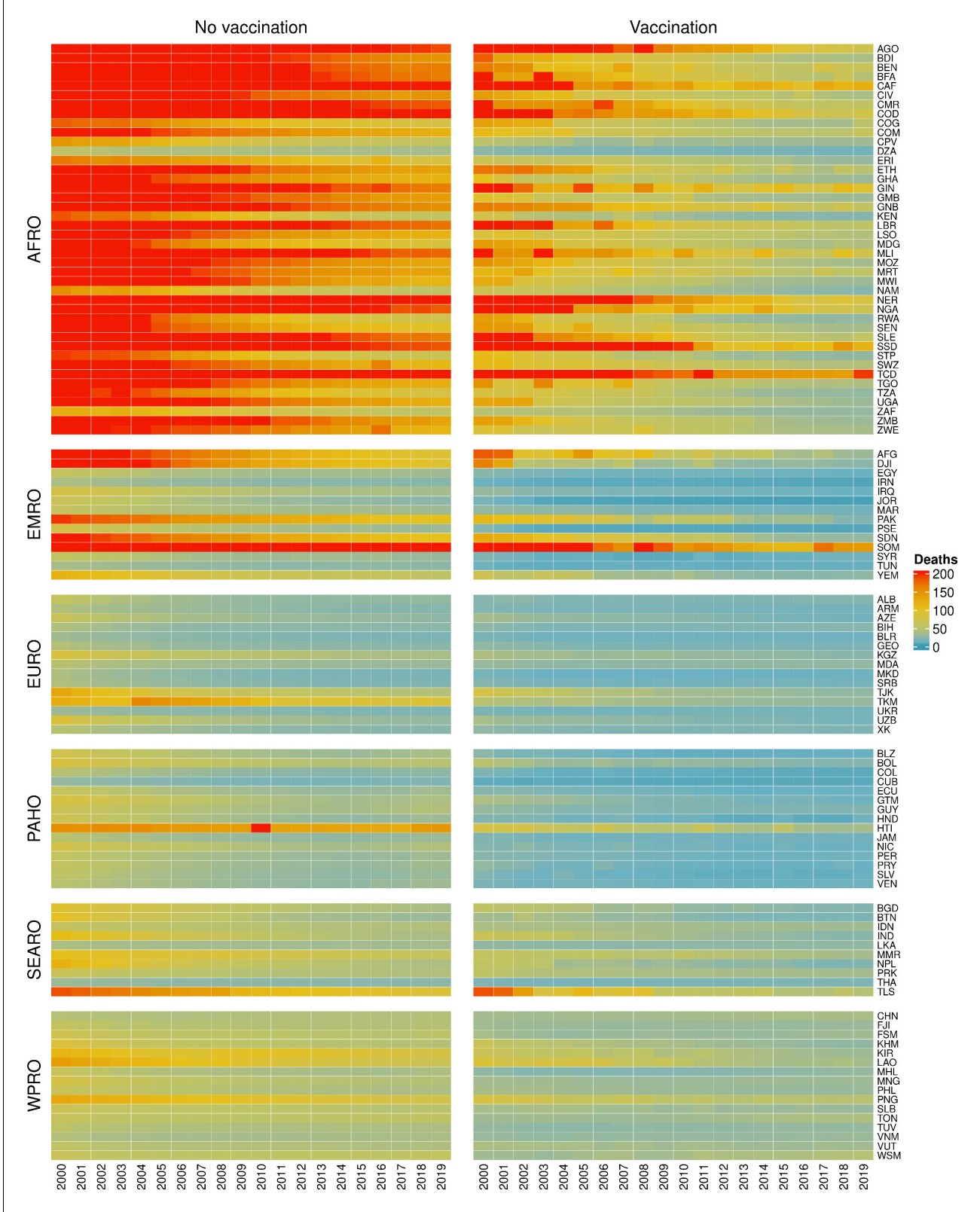

**Figure 1.** Mean predicted deaths due to the 10 Vaccine Impact Modelling Consortium (VIMC) pathogens per 100,000 population per country for years 2000–2019 under the no vaccination and with vaccination (routine immunisations; RI only) scenarios. Countries are arranged by World Health Organisation (WHO) African (AFRO), Eastern Mediterranean (EMRO), European (EURO), Pan American (PAHO), South-East Asian (SEARO), and Western Pacific (WPRO) regions. The difference (i.e. deaths averted) between these two scenarios are shown in *Table 2* and *Figure 2*.

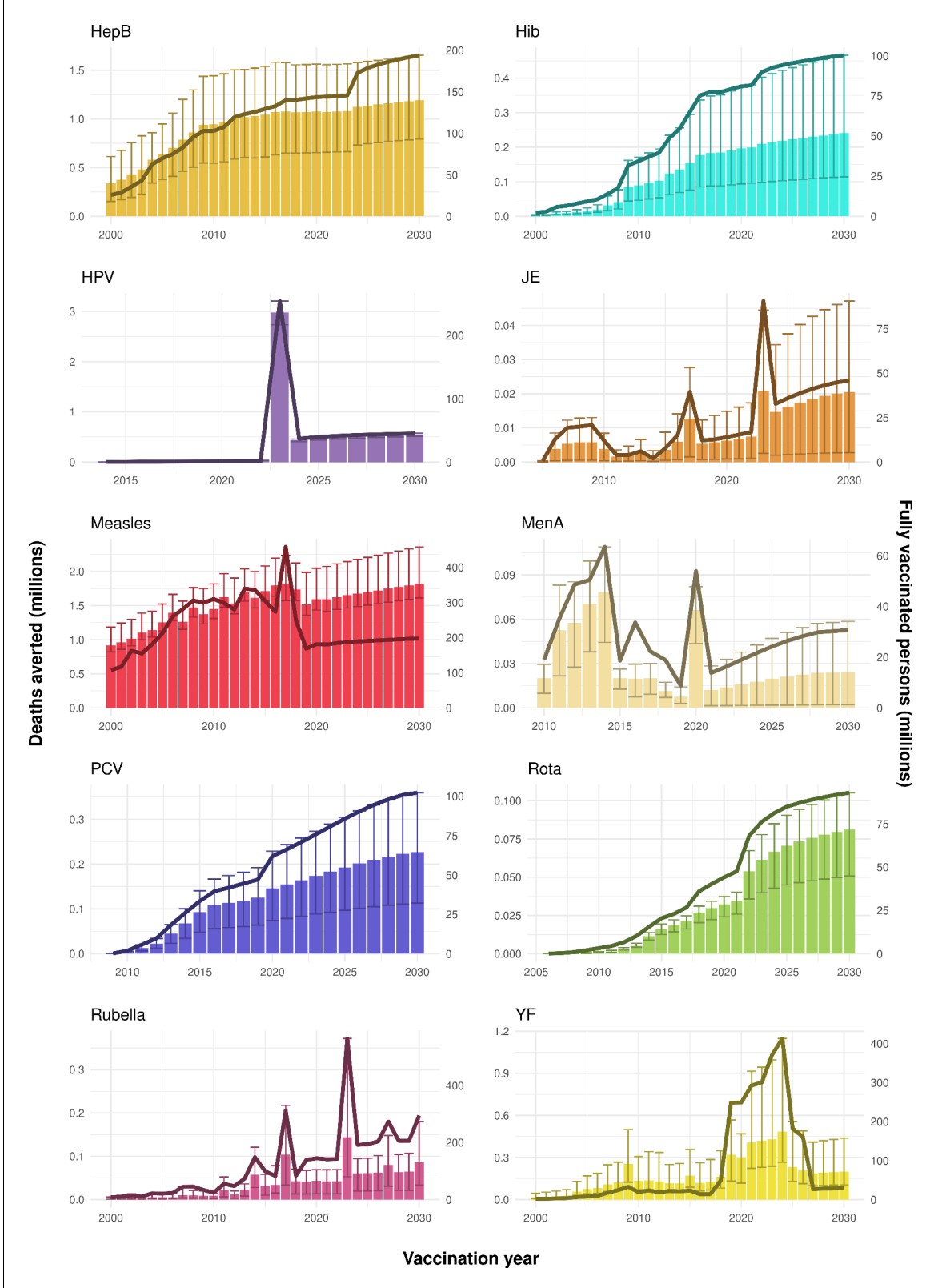

**Figure 2.** Deaths averted per year of vaccination for hepatitis B (HepB), *Haemophilus influenzae* type b (Hib), human papillomavirus (HPV), Japanese encephalitis (JE), measles, *Neisseria meningitidis* serogroup A (MenA), *Streptococcus pneumoniae* (PCV), rotavirus (Rota), rubella, and yellow fever (YF). The bars show the number of deaths averted (in millions) in each vaccination year. Error bars indicate 95% CI. The line shows the number of fully vaccinated persons (FVPs; in millions) achieved in each year's vaccination activities.

**Table 2.** Deaths and disability-adjusted life years (DALYs) averted (in millions), and deaths and DALYs averted per 1000 fully vaccinated people (FVPs) due to vaccination activities in each time period. Numbers within brackets correspond to 95% credible intervals.

| Time period | Deaths averted (in millions) | Deaths averted per 1000 FVPs | DALYs averted (in millions) | DALYs averted per 1000 FVPs |
|---|---|---|---|---|
| 2000–2019 | 50 [41, 62] | 4.8 [3.9, 5.9] | 2700 [2200, 3500] | 260 [210, 330] |
| 2020–2030 | 47 [39, 56] | 3.7 [3.1, 4.4] | 2300 [1900, 2900] | 180 [150, 230] |
| 2000–2030 | 97 [80, 120] | 4.2 [3.5, 5] | 5100 [4100, 6300] | 220 [180, 270] |

than the other pathogens (*Figure 2* and *Appendix 5—figure 3*). Overall, from 2000 to 2030, measles vaccination activities have the largest impact with 47 (95%CrI[42, 60]) million deaths and 3100 (95%CrI[2700, 3900]) million DALYs averted, followed by 29 (95%CrI[17, 43]) million deaths and 1000 (95%CrI[560, 1800]) million DALYs averted due to HepB vaccination activities (*Figure 2* and *Table 3*). Most of the mortality reduction from measles is attributable to routine MCV1, for which procurement is not directly funded by Gavi. As we attribute impact to the year of vaccination, we capture the impact for pathogens where the mortality occurs later in life, such as HepB, whereas, when comparing impact by calendar year (see *Appendix 5—figure 1*), we miss these long-term benefits. As measles-related mortality is focused in children under-5, a large number of DALYs are averted when immunising against this disease. In comparison, as HepB-attributable deaths are primarily focused in those over 40 years of age, there are fewer YLLs but morbidity contributes to higher numbers of YLDs.

Rubella and YF have RI and NRI occurring over the entire time period from 2000 to 2030. With disease burden from CRS modelled for rubella, an estimated 1.2 (95%CrI[0.47, 2.1]) million deaths and 86 (95%CrI[56, 170]) million DALYs are averted. Over the relatively fewer (36) countries endemic for YF, 5.6 (95%CrI[2.9, 13]) million deaths and 210 (95%CrI[110, 510]) million DALYs are estimated to be averted (*Figure 2* and *Table 3*).

For Hib, which is based only on RI, there are an estimated 4.1 (95%CrI[1.9, 7.9]) million deaths and 280 (95%CrI[120, 540]) million DALYs averted over 2000 to 2030 (*Figure 2* and *Table 3*).

Further vaccines for the 10 pathogens have been introduced from 2005 onward, contributing to more lives saved (*Table 1*). From 2014, introductions of RI and NRI for HPV over 112 countries avert an estimated 6.6 (95%CrI[6.1, 7.1]) million deaths and 140 (95%CrI[130, 150]) million DALYs by 2030 (*Figure 2* and *Table 3*). RI were introduced for PCV in 2009, resulting in a further 2.8 (95%CrI[1.4, 4.4]) million deaths and 190 (95%CrI[94, 300]) million DALYs averted by 2030, and for Rota in 2006

**Table 3.** Deaths and disability-adjusted life years (DALYs) averted (in millions), and deaths and DALYs averted per 1000 fully vaccinated people (FVPs) per disease from vaccination activities occurring from 2000 to 2030.

Disease abbreviations: hepatitis B (HepB), human papillomavirus (HPV), yellow fever (YF), *Haemophilus influenzae* type b (Hib), *Streptococcus pneumoniae* (PCV), rotavirus (Rota), *Neisseria meningitidis* serogroup A (MenA), and Japanese encephalitis (JE). Numbers within brackets correspond to 95% credible intervals.

| Disease | Deaths averted (in millions) | Deaths averted per 1000 FVPs | DALYs averted (in millions) | DALYs averted per 1000 FVPs |
|---|---|---|---|---|
| Measles | 47 [42, 60] | 6.5 [5.9, 8.2] | 3100 [2700, 3900] | 420 [380, 540] |
| HepB | 29 [17, 43] | 7.7 [4.7, 12] | 1000 [560, 1800] | 270 [140, 460] |
| HPV | 6.6 [6.1, 7.1] | 12 [11, 13] | 140 [130, 150] | 250 [230, 270] |
| YF | 5.6 [2.9, 13] | 2.1 [1.1, 4.6] | 210 [110, 510] | 81 [43, 200] |
| Hib | 4.1 [1.9, 7.9] | 2.4 [1.1, 4.5] | 280 [120, 540] | 160 [74, 310] |
| PCV | 2.8 [1.4, 4.4] | 2.3 [1.1, 3.7] | 190 [94, 300] | 160 [79, 260] |
| Rubella | 1.2 [0.47, 2.1] | 0.3 [0.1, 0.5] | 86 [56, 170] | 22 [14, 44] |
| Rota | 0.84 [0.56, 1.1] | 0.8 [0.5, 1] | 46 [36, 56] | 44 [35, 54] |
| MenA | 0.62 [0.47, 0.86] | 1 [0.8, 1.4] | 36 [24, 45] | 59 [39, 73] |
| JE | 0.23 [0.03, 0.52] | 0.4 [0, 0.8] | 24 [2.6, 46] | 40 [4.2, 76] |

resulting in 0.84 (95%CrI[0.56, 1.1]) million deaths and 46 (95%CrI[36, 56]) million DALYs averted by 2030 (*Figure 2* and *Table 3*). MenA is endemic in 26 countries with RI and NRI introduced from 2005 onward resulting in 0.62 (95%CrI[0.47, 0.86]) million deaths and 36 (95%CrI[24, 45]) million DALYs averted by 2030 (*Figure 2* and *Table 3*). JE is endemic in fewer (17) countries with RI and NRI also introduced from 2005 onward resulting in 0.23 (95%CrI[0.03, 0.52]) million deaths and 24 (95% CrI[2.6, 46]) million DALYs averted by 2030 (*Figure 2* and *Table 3*).

When examining deaths averted per 1000 FVPs, HPV vaccination activities are estimated to have the largest impact with 12 (95%CrI[11, 13]) deaths averted per 1000 FVPs. This is followed by HepB with 7.7 (95%CrI[4.7, 12]) deaths averted per 1000 FVPs and measles with 6.5 (95%CrI[5.9, 8.2]) deaths averted per 1000 FVPs (*Table 3*). In terms of DALYs averted per 1000 FVPs, measles is estimated to have the largest impact with 420 (95%CrI[380, 540]) DALYs averted per 1000 FVPs as it mainly affects children under-5 (*Table 3*).

Generally, for each of the pathogens, as the number of FVPs (or number of vaccine doses distributed) increase over time, the number of deaths averted increases (*Figure 2*). For the pathogens with RI-only (HepB, Hib, PCV, and Rota), there is an increasing trend of FVPs from 2000 to 2030 leading to a steady increase in deaths averted over this time period. When NRI also occur (HPV, JE, measles, MenA, rubella, and YF), more variation is seen as the FVPs and in turn the deaths averted rise in years for which both activities occur. For example, we expect to see the largest impact due to vaccination activities occurring in the year 2023 for HPV and rubella which project a sharp increase in the number of FVPs arising from NRI in addition to RI in that year (*Figure 2*).

## Impact in children under five

Several of the pathogens, namely Hib, measles, Rota, rubella and PCV, have mortality heavily focused in children under-5. To determine the impact of vaccination for these ages, we aggregate by birth cohort rather than by year of vaccination as this allows us to calculate the disease burden across the first 5 years of life for each yearly birth cohort (born between 2000 and 2030) (*Echeverria-Londono et al., 2021*). We also account for the double counting of mortality when we aggregate mortality across all diseases.

For the 2000–2030 birth cohorts, we estimate that 52 (95%CrI[41, 69]) million deaths and 3400 (95%CrI[2700, 4600]) million DALYs are averted in children under-5. Of these, 33 (95%CrI [27, 43]) million deaths and 2100 (95%CrI[1700, 2800]) million DALYs are estimated to be averted over the years 2000–2019 prior to COVID-19 (*Table 4*). The proportional change due to the removal of double counting is relatively small at 2.36% (95% CI[2.00%, 2.83%]) for all cohorts born between 2000 and 2030 and this reduces to 1.07% (95% CI[0.90%, 1.32%]) for children under-5.

## Impact in comparison to other studies

Our results have focused on the impact by year of vaccination. However, as in the previous VIMC-wide study *Li et al., 2019*, we also investigated the impact of vaccination (deaths and DALYs averted) by calendar year and by birth cohort (*Echeverria-Londono et al., 2021*; *Appendix 5—figure 1* and *Appendix 5—figure 2*). There are differences when comparing the impact estimates, largely driven by changes in coverage/FVPs (*Appendix 5—figure 3*) and/or further developments of model structures, particularly for HepB, HPV, measles and YF. Additional models have also been added, namely, the Emory University Rota model and the University of Notre Dame YF model. Furthermore since the previous study, the uncertainty ranges/confidence intervals for many of the pathogens have narrowed (*Appendix 5—figure 1* and *Appendix 5—figure 2*).

**Table 4.** Deaths and disability-adjusted life years (DALYs) averted (in millions), and deaths and DALYs averted per 1000 fully vaccinated people (FVPs) in children under-5 for birth cohorts born between each time period.

These are adjusted for double counting. Numbers within brackets correspond to 95% credible intervals.

| Time period | Deaths averted (in millions) | Deaths averted per 1000 FVPs | DALYs averted (in millions) | DALYs averted per 1000 FVPs |
|---|---|---|---|---|
| 2000–2019 | 33 [27, 43] | 3.9 [3.2, 5] | 2100 [1700, 2800] | 250 [200, 330] |
| 2020–2030 | 20 [14, 26] | 1.9 [1.3, 2.5] | 1300 [960, 1800] | 130 [95, 180] |
| 2000–2030 | 52 [41, 69] | 2.8 [2.2, 3.7] | 3400 [2700, 4600] | 190 [140, 250] |

Mortality estimates from our results were compared to the IHME GBD 2019 (*Institute for Health Metrics and Evaluation, 2019*) on a global level and for four high burden countries (Pakistan, India, Nigeria and Ethiopia) for HepB, measles and YF. Note, estimates for GBD 2019 are global and for VIMC are for 112 countries. The GBD 2019 did not estimate deaths averted. Comparison between the mortality estimates from VIMC and GBD 2019 show significant overlap in the overall values between 2000 and 2019 for HepB and measles (*Table 5*). Globally, measles mortality estimates from VIMC tend to be higher than those from GBD 2019 between 2000 and 2010 with an increasing overlap in recent years (*Appendix 5—figure 8*). For HepB, the trend is reversed with overlapping estimates between 2000 and 2010 and divergent estimates in recent years (*Appendix 5—figure 6*). For measles, VIMC has greater variability in the mortality estimates in countries with a high burden such as Pakistan, India, Nigeria, and Ethiopia compared to GBD 2019 estimates (*Appendix 5—figure 9*). For HepB, we see considerable agreement between the VIMC and GBD 2019 mortality in Pakistan, India and Nigeria (*Appendix 5—figure 7*). Unlike measles and HepB, the global mortality estimates for YF from VIMC do not show any overlap with those from GBD 2019, with significantly higher VIMC estimates (*Table 5* and *Appendix 5—figure 10*). Nevertheless, when looking at the mortality estimates for a high burden country such as Ethiopia, we do see overlap between the estimates but with great uncertainty (*Appendix 5—figure 11*). The differences between VIMC and GBD 2019 estimates are generally due to differences in treatment assumptions and parameter values, such as the CFR estimates for YF.

## Discussion

We present the first estimates of vaccine impact to be attributed to the year in which the vaccination activity occurred for 10 pathogens in 112 countries. This alternative view of impact allows us to directly assess the influence of a particular year's vaccination efforts over the lifespans of all individuals affected, better capturing the full long-term benefits of vaccination. This is an advance both in methodology and scope with the countries and pathogens considered representing the vast majority of VPD burden globally.

Stratifying the impact of vaccination activities over the years 2000–2019 and 2020–2030 allows us to estimate the immense progress made to date, and to estimate future advances which may be affected by the COVID-19 pandemic as well as other variations in transmission or healthcare. Without vaccination activities between 2000 and 2019, there would be an additional 50 (95%CrI[41, 62]) million deaths, with a further additional 47 (95%CrI[39, 56]) million deaths without vaccination activities between 2020 and 2030 due to these 10 pathogens over the 112 countries. If vaccination proceeds per the default scenario forecast through 2030, the greatest reductions in deaths are predicted to be for measles with 47 (95%CrI[42, 60]) million deaths averted from vaccination activities occurring in 2000 to 2030. HepB, HPV and YF also see large predicted reductions with 29 (95% CrI[17, 43]), 6.6 (95%CrI[6.1, 7.1]) and 5.6 (95%CrI[2.9, 13]) million deaths averted, respectively. In children under-5, we examine the impact per birth cohort and find that an estimated 33 (95%CrI[27,

**Table 5.** Global mortality estimates (in thousands) from the Vaccine Impact Modelling Consortium (VIMC) and the Global Burden of Disease Study (GBD) 2019 from the Institute for Health Metrics and Evaluation (IHME) attributed to Hepatitis B (HepB), measles and yellow fever (YF) for all ages and for children under-5 over the years 2000–2019.
Estimates for GBD 2019 are global and for VIMC are for 112 countries. 95% CI shown for VIMC estimates (see *Appendix 5—figures 6–11*).

| Disease | Time period | All ages | | Under-5 | |
|---|---|---|---|---|---|
| | | VIMC 2019 | GBD 2019 | VIMC 2019 | GBD 2019 |
| HepB | 2000–2010 | 7200 [5100, 10000] | 5200 | 100 [21, 360] | 72 |
| | 2011–2019 | 7200 [5300, 9800] | 4200 | 33 [5.4, 110] | 43 |
| Measles | 2000–2010 | 5600 [4100, 9500] | 4200 | 5300 [3800, 9400] | 3600 |
| | 2011–2019 | 920 [620, 1700] | 1200 | 870 [560, 1700] | 1100 |
| YF | 2000–2010 | 600 [320, 1500] | 84 | 100 [54, 250] | 10 |
| | 2011–2019 | 450 [240, 1100] | 47 | 63 [32, 150] | 5.7 |

43]) million child lives were saved by vaccination between 2000 and 2019, 20 (95%CrI[14, 26]) million thereafter.

In comparison to other studies, we generally find less uncertainty and lower median deaths averted estimates relative to the previous VIMC-wide study (*Li et al., 2019*) and similar overall mortality estimates to the IHME GBD 2019 (*Institute for Health Metrics and Evaluation, 2019*). The differences compared to the previous VIMC-wide study which examined the same pathogens but for a subset of countries (98 of the 112 countries), are mostly driven through differing assumptions around FVPs, affecting HPV, developments to model structure which influence the results for HepB, measles and YF, and additional models for Rota and YF. In comparison to GBD 2019 for HepB and measles, we find similar magnitude estimates of mortality both globally and for particular high-burden countries (Nigeria, Pakistan, India, and Ethiopia). However, YF estimates diverge from the GBD 2019 due to differences in assumed CFR values and parameter estimates.

In this study, we attribute the impact to the year in which the vaccination activity occurred through calculating an impact ratio. We stratify this impact ratio by vaccine activity type (assuming vaccine impact does not vary between birth cohorts), thereby averaging the effects of any improvements in disease incidence or population health over the entire time period modelled. However, a recent study examined different ratio stratifications and found varying support for each dependant on the question, pathogen and model examined (*Echeverria-Londono et al., 2021*). As such, whilst we have shown the impact in one format, this could underestimate characteristics such as the change in population demography, transmission or healthcare over time which may mean that one cohort has a different experience of vaccination compared to another. Similarly, the assumptions around vaccination post-2030 may have implications for the impact of earlier activities, for example in rubella the number of CRS cases depend on infections among women of child-bearing age, thus later vaccination activities could affect the incidence of CRS over the lifetime of vaccinated individuals. As a result, for long-term disease burden due to pathogens such as HepB or HPV, we may underestimate the uncertainty in vaccine impact. This may be particularly relevant when assessing the changes in healthcare due to COVID-19 and the introduction of SARS-COV-2 vaccines.

We account for uncertainty in model structure and parameterisation by including at least two models per pathogen sampling from the full uncertainty distribution of both models. However, we do not consider uncertainties within demography or immunisation coverage data. Demographic uncertainty will affect both our estimates of vaccination coverage and the disease dynamics themselves. For example, although the UNWPP population data takes migration into account, we do not account for this explicitly. As such, we may lose a key influencing factor for disease transmission from one country to another. Furthermore, estimating current vaccine uptake is often complicated by changes in assumed population size and issues around dose wastage. This is one reason that the coverage estimates for RI and NRI are uncertain, affecting any measure of vaccine impact. The correlation and dependency between doses varies by disease modelled and can influence the relative effects of campaign versus routine immunisation. Although vaccines such as measles and rubella may be given together, we considered them to be independent.

Inclusion of different model structures allows us to capture some of the inherent unknowns within the epidemiology of these pathogens. However, in some cases, data are limited and validation of models is not possible. This is a focus of constant work as more data becomes available. Conversely, as we focus on 112 countries, a limitation of our study is that not all countries are modelled globally. However, our analyses include the countries with the highest burden relating to the pathogens (representing 99% of the total mortality attributed to measles for under-5s [*World Health Organization, 2020*] and 96% of the total deaths attributed to HepB, Hib, measles, MenA, PCV, and YF of all ages [*Institute for Health Metrics and Evaluation, 2019*]).

Following vaccine introductions, future coverage has been assumed to increase over time. However, there is the risk of decline in coverage, or delays to activities without sustained focus. Disruptions to health services caused by the COVID-19 pandemic have been an example of such disruption and in April 2020, Gavi estimated that at least 13.5 million people may have missed vaccinations with disruption likely to continue (*Gavi, the Vaccine Alliance, 2020*). Similarly, *Chandir et al., 2020* estimated that one in every two children in Sindh province, Pakistan have missed their routine vaccinations during lockdown associated with the pandemic. Disruption to vaccine and health care services may influence our estimates of lives saved from 2020 onward, particularly if the risk of outbreak or disease emergence is increased. However, this disruption in immunisation might be partially offset

by decreased disease transmission due to NPIs implemented to help control COVID-19, as has been shown for influenza and norovirus (*Jones, 2020*; *Kraay et al., 2020*). In the longer-term, there is a risk that NPIs may result in a build up of susceptible individuals in the population for outbreak prone diseases, such as measles, but catch-up activities may be able to prevent this. To date, many vaccination activities have restarted and catch-up vaccination campaigns have begun to ensure the immunity gap due to disruption is as small as possible.

Despite improvements in vaccine coverage, universal vaccination coverage is not yet achieved and there are areas in many countries where coverage remains low (*World Health Organization, 2021b*; *Hamlet et al., 2019*; *Kundrick et al., 2018*; *Takahashi et al., 2017*; *Vanderslott et al., 2013*). The model estimates presented in this study do not account for such geographic or socioeconomic clustering of vaccine coverage, which could increase disease transmission. Hence, sub-populations with low access to vaccines and/or high exposure to the pathogens are not presented in our results (*Gavi, the Vaccine Alliance, 2020*; *Chandir et al., 2020*). However, some of the models included are estimated subnationally and can examine questions around heterogeneity in health access and transmission (Appendix 2.2). A combined, cross-pathogen approach to these heterogeneities is an area of continued research.

When attributing vaccine impact to the year of vaccination, and aggregating mortality across all 10 pathogens, we do not adjust for double counting, thereby counting an individual's death more than once when mortality arises by more than one pathogen (*Li et al., 2019*). However, this is accounted for when aggregating vaccine impact over a calendar year and birth cohort. The issue of double counting can be viewed from two perspectives- either a person's life is saved from different pathogens multiple times or their death is averted from different pathogens multiple times. Intuitively, the former makes sense, it is important to capture each time an individual's life is saved. The latter is a more difficult perspective as each person will only die once. When focusing on the under-5s using the birth cohort method, the proportional change due to double counting adjustment was found to be 2.36% (95% CI[2.00%, 2.83%]) for cohorts born between 2000 and 2030 and reduced to 1.07% (95% CI[0.90%, 1.32%]) for under-5s. Thus, whilst the majority of double counting occurs in the under-5s, the overall difference is minimal.

Although we do not account for the current COVID-19 pandemic, our analyses provide a vital baseline against which comparison can be made. Studies assessing the impact of COVID-19 on VPDs are ongoing. *Abbas et al., 2020a* assessed the benefit of continuing routine childhood immunisation in Africa given the ongoing pandemic. They found the benefits outweighed the costs with 84 (95% uncertainty interval 14-267) child deaths averted by sustained childhood immunisation per 1 excess COVID-19 death even with the risks associated with vaccination clinic visits. The VIMC Working Group on COVID-19 Impact on VPDs analysed the effect of COVID-19 disruption on measles, MenA and YF through modelling scenarios of routine immunisation service disruptions and mass vaccination campaign suspensions in a subset of countries (*Gaythorpe et al., 2021b*). They found that the nature of the disease affects the impact of vaccination activity disruption; for example, YF and measles affect younger age groups and are prone to outbreaks, thus short-term disruption will likely increase burden. However, protection afforded by previous vaccination activities for MenA can mitigate the short-term effects due to COVID-19 disruption. A global analysis of the impact of COVID-19 on vaccination activities is not yet available and it is unclear how the continued disruption, and likely impact of distributing a future SARS-COV-2 vaccine, will affect vaccination in the future. Conversely, we also do not know to what extent transmission has been perturbed due to NPIs instigated to mitigate COVID-19 for the pathogens mentioned here.

Overall, our results provide a thorough assessment of the impact of vaccination activities prior to COVID-19, from 2000 to 2019, and from 2020 thereafter. These results are subject to change as our understanding of the transmission and epidemiology of these pathogens continues to grow. Additionally, future coverage, particularly during and following the pandemic, is uncertain. This study paints a picture of the immense progress to date and the tremendous health impacts that could be obtained over the next decade due to vaccination activities.

## Conclusion

Our largest VIMC-wide study for 10 pathogens across 112 countries showcases the immense impact of vaccination activities over 2000–2030 with 97 (95%CrI[80, 120]) million lives estimated to be saved in a pre-COVID-19 world. Though the wide-spread COVID-19 pandemic has caused disruption to

vaccination activities, currently it is difficult to assess the impact. Nonetheless, our study shows the substantial progress to date and as we look to the future, it continues to show the benefits of vaccination and motivates efforts to sustain and improve coverage of vaccination globally.

## Acknowledgements

We acknowledge William Msemburi for preparing forecasts of coverage from 2019 to 2030 as described in the supplementary material. We would also like to thank Rob Ashton, Rich FitzJohn, Alex Hill, Emma Russell and Mark Woodbridge for their technical support, and Diana O'Malley for support with project coordination.

## Additional information

### Competing interests

Mark Jit: Reviewing editor, *eLife*. Michael L Jackson: MLJ has received research funding from Sanofi Pasteur unrelated to the present work. Benjamin A Lopman: BAL reports grants and personal fees from Takeda Pharmaceuticals, personal fees from World Health Organization, outside the submitted work. T Alex Perkins: TAP receives support from Emergent Biosolutions for work unrelated to his contribution to this study. Homie Razavi: HR is an employee of Center for Disease Analysis Foundation which has received grants from Gilead Sciences, AbbVie, Zeshan Foundation and EndHep2030 fund for projects unrelated to this work; HBV epidemiology data was funded by a grant from John Martin Foundation (Grant number 24). Caroline L Trotter: CLT received a consulting payment from GSK in 2018 (outside the submitted work). The other authors declare that no competing interests exist.

### Funding

| Funder | Grant reference number | Author |
|---|---|---|
| Bill and Melinda Gates Foundation | OPP1157270 / INV-009125 | Jaspreet Toor<br>Susy Echeverria-Londono<br>Xiang Li<br>Kaja Abbas<br>Emily D Carter<br>Hannah E Clapham<br>Andrew Clark<br>Margaret J de Villiers<br>Kirsten Eilertson<br>Matthew Ferrari<br>Ivane Gamkrelidze<br>Timothy B Hallett<br>Wes R Hinsley<br>Daniel Hogan<br>John H Huber<br>Michael L Jackson<br>Kevin Jean<br>Mark Jit<br>Andromachi Karachaliou<br>Petra Klepac<br>Alicia Kraay<br>Justin Lessler<br>Xi Li<br>Benjamin A Lopman<br>Tewodaj Mengistu<br>C Jessica E Metcalf<br>Sean M Moore<br>Shevanthi Nayagam<br>Timos Papadopoulos<br>T Alex Perkins<br>Allison Portnoy<br>Homie Razavi<br>Devin Razavi-Shearer<br>Stephen Resch<br>Colin Sanderson<br>Steven Sweet |

| | | Yvonne Tam |
| | | Hira Tanvir |
| | | Quan Tran Minh |
| | | Caroline L Trotter |
| | | Shaun A Truelove |
| | | Emilia Vynnycky |
| | | Neff Walker |
| | | Amy Winter |
| | | Kim Woodruff |
| | | Neil M Ferguson |
| Medical Research Council | MR/R015600/1 | Jaspreet Toor |
| | | Susy Echeverria-Londono |
| | | Xiang Li |
| | | Wes R Hinsley |
| | | Kim Woodruff |
| | | Neil M Ferguson |
| | | Katy AM Gaythorpe |
| National Institutes of Health | R01 GM124280 | Benjamin A Lopman |
| National Institutes of Health | R01 AI112970 | Benjamin A Lopman |

This publication is authored by members of the Vaccine Impact Modelling Consortium (VIMC, www.vaccineimpact.org). VIMC is jointly funded by Gavi, the Vaccine Alliance, and by the Bill Melinda Gates Foundation. The views expressed are those of the authors and not necessarily those of the Consortium or its funders. The funders were given the opportunity to review this paper prior to publication, but the final decision on the content of the publication was taken by the authors. Consortium members received funding from Gavi and BMGF via VIMC during the course of the study.

## Author contributions

Jaspreet Toor, Susy Echeverria-Londono, Xiang Li, Conceptualization, Formal analysis, Investigation, Methodology, Writing - original draft, Writing - review and editing; Kaja Abbas, Conceptualization, Software, Formal analysis, Investigation, Methodology, Writing - review and editing; Emily D Carter, Hannah E Clapham, Petra Klepac, Xi Li, Steven Sweet, Shaun A Truelove, Formal analysis, Writing - review and editing; Andrew Clark, Formal analysis, Methodology, Writing - review and editing; Margaret J de Villiers, Kevin Jean, Sean M Moore, Devin Razavi-Shearer, Stephen Resch, Data curation, Formal analysis, Writing - review and editing; Kirsten Eilertson, Methodology; Matthew Ferrari, Quan Tran Minh, Formal analysis, Methodology; Ivane Gamkrelidze, Conceptualization, Formal analysis, Methodology, Writing - review and editing; Timothy B Hallett, Supervision, Investigation; Wes R Hinsley, Data curation, Software; Daniel Hogan, Conceptualization, Data curation, Funding acquisition, Validation, Investigation, Writing - review and editing; John H Huber, Software, Investigation, Methodology, Writing - review and editing; Michael L Jackson, Conceptualization, Formal analysis, Investigation, Visualization, Methodology, Project administration, Writing - review and editing; Mark Jit, Data curation, Formal analysis, Validation, Investigation, Methodology, Writing - review and editing; Andromachi Karachaliou, Conceptualization, Data curation, Formal analysis, Validation, Methodology, Writing - review and editing; Alicia Kraay, Formal analysis, Investigation, Methodology, Writing - review and editing; Justin Lessler, Conceptualization, Resources, Formal analysis, Supervision, Investigation; Benjamin A Lopman, Methodology, Writing - review and editing; Tewodaj Mengistu, Resources, Data curation, Validation, Investigation, Writing - review and editing; C Jessica E Metcalf, Conceptualization, Supervision, Methodology; Shevanthi Nayagam, Supervision, Writing - review and editing; Timos Papadopoulos, Data curation, Software, Validation, Visualization, Methodology, Writing - review and editing; T Alex Perkins, Resources, Investigation, Writing - review and editing; Allison Portnoy, Data curation, Writing - review and editing; Homie Razavi, Data curation, Validation; Colin Sanderson, Data curation, Methodology, Writing - review and editing; Yvonne Tam, Software, Formal analysis, Writing - review and editing; Hira Tanvir, Formal analysis, Investigation; Caroline L Trotter, Formal analysis, Supervision, Investigation, Methodology, Writing - review and editing; Emilia Vynnycky, Software, Supervision, Writing - review and editing; Neff Walker, Formal analysis, Supervision; Amy Winter, Formal analysis, Validation, Methodology; Kim Woodruff, Project administration, Writing - review and editing; Neil M Ferguson, Conceptualization, Methodology,

Writing - review and editing; Katy AM Gaythorpe, Conceptualization, Data curation, Formal analysis, Validation, Investigation, Visualization, Methodology, Writing - original draft, Writing - review and editing

## Author ORCIDs
Jaspreet Toor https://orcid.org/0000-0003-1510-397X
Hannah E Clapham https://orcid.org/0000-0002-2531-161X
Kevin Jean https://orcid.org/0000-0001-6462-7185
Mark Jit http://orcid.org/0000-0001-6658-8255
Petra Klepac http://orcid.org/0000-0003-4132-3933
Justin Lessler http://orcid.org/0000-0002-9741-8109
Tewodaj Mengistu http://orcid.org/0000-0003-3475-3599
C Jessica E Metcalf http://orcid.org/0000-0003-3166-7521
Sean M Moore http://orcid.org/0000-0001-9062-6100
T Alex Perkins https://orcid.org/0000-0002-7518-4014
Shaun A Truelove http://orcid.org/0000-0003-0538-0607
Kim Woodruff https://orcid.org/0000-0003-4618-8267
Katy AM Gaythorpe https://orcid.org/0000-0003-3734-9081

## Decision letter and Author response
Decision letter https://doi.org/10.7554/eLife.67635.sa1

# Additional files

## Supplementary files
• Transparent reporting form

## Data availability

Data is available through a publicly available tool at https://montagu.vaccineimpact.org/2021/visualisation/.

The following dataset was generated:

| Author(s) | Year | Dataset title | Dataset URL | Database and Identifier |
|---|---|---|---|---|
| Toor J, Echeverria-Londono S, Xiang L, Abbas K, Carter ED, Clapham H, Clark A, De Villiers MJ, Eilertson K, Ferrari M, Gamkrelidze I, Hallett TB, Hinsley WR, Hogan D, Huber JH, Jackson ML, Jean K, Jit M, Karachaliou A, Klepac P, Kraay A, Lessler J, Li X, Lopman BA, Mengistu T, Metcalf CJE, Moore SM, Nayagam S, Papadopoulos T, Perkins TA, Portnoy A, Razavi H, Razavi-Shearer D, Resch S, Sanderson C, Sweet S, Tam Y, Tanvir H, Tran Minh Q, Trotter CL, Truelove | 2021 | Estimates of health impact from vaccination against 10 pathogens in 112 low and middle income countries from 2000 to 2030. | https://montagu.vaccineimpact.org/2021/visualisation/ | VIMC Data Visualisation Tool, 2021/visualisation |

SA, Vynnycky E,
Walker N, Winter A,
Woodruff K,
Ferguson NM,
Gaythorpe KA

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

# Appendix 1

## Vaccine coverage forecasts

Vaccine coverage forecasts were generated for every country-antigen combination for years 2019 to 2030. For existing routine programs, future coverage was modelled using Generalised Additive Models (GAMS), with historical WUENIC coverage as an input, to forecast the country- and antigen-specific time-series of coverage (*WHO UNICEF coverage estimates, 2020*).

For countries that had not yet introduced a specific vaccine by 2018, all countries were assigned the same future year of introduction for that vaccine based on projections from an accelerated failure time model. The vaccine introduction year and coverage were forecasted for each vaccine based on the association between time to introduction and coverage of existing vaccines (DTP3, MCV1 and Pol3) in countries where the vaccine had been introduced, controlling for Gavi support. Coverage scale-up was then modeled for each country on the basis of annual rates of increase in coverage for existing vaccines at the time of their introduction. For regional vaccines, namely yellow fever (YFV), meningitis A (MenA) and Japanese Encephalitis (JE), vaccine introductions were forecasted only in countries where the respective disease is endemic.

Non-routine activities, such as supplementary Immunisation activities (SIAs) and multi-age cohort vaccinations (MACs), and their associated fully immunised persons (FVPs) were forecasted for the regional vaccines, measles/measles-rubella (M/MR) and HPV, according to the specific characteristics of the individual vaccine program and the related WHO recommendations, see table *Appendix 1—table 1* for summary.

**Appendix 1—table 1.** Vaccine, non-routine immunisation activity type and target population for coverage forecasts.

| Vaccine | Type of non-routine immunisation activity | Target population |
|---------|-------------------------------------------|-------------------|
| HPV | Multi-age cohorts (MACs) | Girls 10–14 year-olds |
| MR | Catch-up | 9 months- 15 year-olds |
| M/MR | Follow-up | 9 months- 5 year-olds |
| JE | Catch-up | 9 months −14 year-olds |
| Men A | Initial catch-up | 1–29 year-olds |
| | Mini-catch-up | Country dependent |
| YFV | Preventive mass campaigns | At risk population older than 9 months |

For MR, HPV, and JE, introductions of the vaccines in the routine program are generally preceded with catch-up campaigns or MACs for HPV. Thus, the analysis outlined above also predicts campaigns as they relate to the routine introductions, according to the specifications outlined in table below.

For Men A, the WHO recommends an initial introductory campaign targeting 1 to 29 year-olds and a mini catch-up campaign targeting all the missed cohorts between the initial catch-up and the introduction of the vaccine into routine. Thus, in countries where introductory campaigns had been conducted by 2018, mini catch-up campaigns were forecasted at the time of routine introduction, and the target populations were calculated as the number of cohorts missed between the initial introductory campaign and the routine introduction date. Where introductory campaigns had not been conducted by 2018, we assumed these would be conducted in 2019.

Measles and measles-rubella (M/MR) follow-up campaigns were forecasted on the basis of WHO guidance on the frequency of these types of campaigns, that is, every 2 years in countries where MCV1 coverage is less than 60%, every 3 years in countries where MCV1 coverage is between 60% and 79% , and every 4 years in countries where MCV1 coverage is over 80%.

National mass preventive campaigns for yellow fever were included in the forecasts for relevant countries according to proposed sequencing in the WHO EYE Strategy Eye (*Weekly Epidemiol Record, 2017*).

## Appendix 2

## Model review process and model descriptions

### 2.1 Model review process

All VIMC models were reviewed against pre-defined model standards in early 2018. Three pre-defined minimum standards and seven desirable standards set out the criteria for models' inclusion in VIMC. The three essential criteria are (1) the models can produce required outputs: deaths, cases and DALYs for all countries and years; (2) the models used standardised demography data provided by the VIMC and (3) the models are well documented. The seven desirable criteria were: (1) rigorous fitting to epidemiological data; (2) appropriate model complexity for the data available; (3) suitable data used for model fitting; (4) out-of-sample validation; (5) ability to capture quantifiable uncertainty; (6) representation of indirect effects of vaccination (herd immunity) where epidemiologically relevant; (7) shared model source code. The 2018 reviews were led by the VIMC management group. These reviews have been repeated annually against the same standards, but with a move towards light-touch peer reviews.

### 2.2 Model descriptions

### Hepatitis B – Centre for Disease Analysis Foundation

PRoGReSs is a deterministic, dynamic disease burden model of HBV infection that calculates the annual HBV prevalence, incidence, and mortality by stage of liver disease, serologic status (low-viral load [LVL], high-viral load [HVL], on-treatment), sex, and age. A detailed description of the model has been previously published (*Polaris Observatory Collaborators, 2018*). Since the last major publication specific to the model, the following progression rates have been revised: chronic hepatitis B (CHB) to hepatocellular carcinoma (HCC), LVL; CHB to HCC, HVL; compensated cirrhosis (CC) to HCC, LVL; CC to HCC, HVL; HCC to HBV-related death, subsequent years; decompensated cirrhosis (DCC) to HBV-related death; CHB to CC, LVL; CHB to CC, HVL. The remaining progression rates have been previously published (*Polaris Observatory Collaborators, 2018*). The HBV-related deaths were calculated by applying stage-, sex-, and age-specific mortality rates to the HBV-infected population. Updated prevalence estimates since the last publication were incorporated in the analysis (*United States Ministry of Health, 2018*; *Ministry of Health, 2018*; *Quaglio et al., 2008*; *Shrestha, 1990*; *Thompson et al., 2019*; *Tshering et al., 2020*; *UPHIA, 2017*).

Uncertainty in a wide range of model inputs, detailed below, was modeled to estimate the uncertainty in all model outputs. The parameters of the probability distributions were obtained from previously reported sources (*Polaris Observatory Collaborators, 2018*).

The share of HBeAg-negative cases among HVL cases, share of HBeAg-positive cases among HVL cases, and HBV transmission rates to infants born to mothers with HVL with and without peripartum antiviral treatment, having received (1) complete HBV vaccine series with timely dose without HBIG, (2) complete HBV vaccine series with timely birth dose with HBIG, (3) timely birth dose of HBV vaccine only, (4) complete HBV vaccine series without timely birth dose, and (5) no vaccination were assumed to be betaPERT-distributed.

Progression rates of (1) CHB to CC, LVL, (2) CHB to HCC, LVL, (3) CHB to CC, HVL, (4) CHB to HCC, HVL, (5) CC to HCC, LVL, (6) CC to HCC, HVL were parametrized using a random variable corresponding to a (1) baseline (50% likelihood), (2) low (25% likelihood), and (3) high (25% likelihood) progression. Progression rates of (1) CC to DCC, HVL, (2) development of fulminant HBV, (3) DCC to HBV-related death, LVL, (4) mortality from fulminant HBV, (5) HCC to HBV-related death, first year, and (6) HCC to HBV-related death, subsequent years were parametrized using betaPERT-distributed scalar multipliers.

Additionally, the treatment schedule was parametrized as a random variable corresponding to (1) baseline (50% likelihood), (2) intermediate (25% likelihood), and (3) optimistic (25% likelihood) treatment coverages.

Lastly, the incidence function was parametrized with a betaPERT-distributed scalar multiplier corresponding to baseline, low, and high prevalence estimates of HBsAg by country. A Monte Carlo simulation with 200 realizations was performed to calculate uncertainty intervals around all modeled outcomes. No correlations between doses of HBV vaccine were assumed.

## Hepatitis B – Imperial College London

This population-level, deterministic, dynamic transmission model (*Nayagam et al., 2016*; *de Villiers et al., 2020*) contains both acute (Severe Acute and Non-severe Acute) and chronic (Immune Tolerant, Immune Reactive, Asymptomatic Carrier, Chronic Hepatitis B, Compensated Cirrhosis, Decompensated Cirrhosis and Liver Cancer) mutually exclusive disease states. The two acute states as well as the Immune Tolerant and the Immune Reactive states are assumed to contain HBsAg+ HBeAg+ individuals, which are assumed to be 15 times more infectious than the HBsAg+ HBeAg- individuals in the other states (*Mendy et al., 1999*; *Mendy et al., 2008*; *Keane et al., 2016*). The model also contains separate state variables for susceptible and recovered/vaccinated individuals as well as for individuals on tenofovir antiviral treatment. HBV-related deaths can occur from the Severe Acute, Compensated Cirrhosis, Decompensated Cirrhosis, Liver Cancer and treatment states. The rates of progression through disease stages are informed by literature reviews and assumed to be the same in all settings.

Infection is spread in the population by both vertical and horizontal transmission, the rates of which are informed through fitting. The risk of acute infection becoming chronic is highest in the younger age groups and controlled by an exponential function that ranges in value from 88·5% for vertical infections in infants to less than 5% risk for acute sufferers over 30 years of age (*Edmunds et al., 1993*). Background mortality and migration are applied equally to individuals in all states. Younger age groups are assumed to undergo seroconversion from being HBsAg+ HBeAg+ to HBsAg+ HBeAg- at a faster rate than older age groups. In contrast, older age groups are at greater risk of developing liver cancer than are younger age groups. Since new HBV cases predominantly occur among the younger age groups (infants and 1 to 5 year olds), population make-up and fertility rates heavily influence the rate of spread of the disease in the population.

The model takes into account the effects of the birth-dose (BD) and the infant vaccines. The BD vaccine is assumed to be 95% effective in protecting infants of mothers that are HBsAg+ HBeAg-, and 83% effective in protecting infants of mothers that are HBsAg+ HBeAg+. All infants are equally likely to be given the BD vaccine within 24 hr of birth. The infant vaccine is assumed to be 95% effective in conferring life-long protection to vaccinated individuals. Individuals are assumed to be either unvaccinated or have been given all infant vaccine doses necessary in their first six months of life to confer the full protection specified in the model. All six-month-olds are equally likely to be given the infant vaccine series.

The model is calibrated to country-level, age-specific HBsAg+ prevalence data and HBeAg+/ HBsAg+ prevalence data in pregnant women, obtained from the Polaris Observatory and from other sources (*Ott et al., 2012a*; *Ott et al., 2012b*), as well as to HBV-related cirrhosis and liver cancer death rates, obtained from the Global Burden of Disease Results Tool website (*Institute for Health Metrics and Evaluation, 2019*). The calibrated model parameters include the risk of horizontal transmission to susceptible one to five year-olds and the risk of vertical transmission from HBsAg+ HBeAg- mothers to their infants. Calibration is performed by the Approximate Bayesian Computation Sequential Monte Carlo algorithm (*Toni et al., 2009*).

Other data sources include demographic data (female and male population sizes of one-year age groups, migration, female fertility rates of five-year age groups, sex ratio of infants, female and male life expectancy of five-year age groups, female and male mortality rates of five-year age groups) from the United Nations World Population Prospects (UNWPP) 2019 Revision and infant and BD vaccine coverage data from the WHO/UNICEF Estimates of National Immunization Coverage (WUENIC) and Gavi, the Vaccine Alliance.

Uncertainty in model estimates are due to uncertainties in the prevalence data that are used in the model calibration, as well as uncertainties in historical coverage data, vaccine efficacies and the number of individuals on antiviral treatment. The model structure is regularly updated to reflect the latest understanding of the natural history of HBV.

## Hepatitis B – Goldstein

The model was developed by Susan Goldstein, Fangjun Zhou, Stephen Hadler, Beth Bell, Eric Mast and Harold Margolis at the US Centers for Disease Control and Prevention (CDC) (*Goldstein et al.,*

*2005*). It is a static deterministic model that estimates the global burden of hepatitis B and the impact of hepatitis B immunization programs. The model examines the mortality outcomes due to hepatitis B virus (HBV) infection, including deaths of fulminant hepatitis, and deaths of liver cirrhosis and hepatocellular carcinoma as results of chronic hepatitis B.

The model assumes infections occur in three age periods with different probabilities of developing symptomatic infections and progressing to chronic hepatitis B, which are: perinatal period, early childhood period (under 5 years), and the period over 5 years of age.

The rate of perinatal infection was determined by the prevalence of hepatitis B surface antigen (HBsAg) and hepatitis B e antigen (HBeAg) among pregnant women. Infants born to HBsAg positive and HBeAg positive mothers had a 90% chance of perinatal infection, while infants born to HBsAg positive and HBeAg negative mothers had a 10% chance of perinatal infection. The rate of infection in early childhood was determined by the prevalence of antibody to hepatitis B core antigen (anti-HBc) at age five after excluding perinatal infections, and the rate of infections between age 5 and 30 was determined by anti-HBc prevalence at 5 and 30 years of age. The prevalence at 30 years of age was assumed to have reached its peak in lifetime. A literature review was conducted on the prevalence of the hepatitis B seromarkers worldwide, and countries were grouped into 15 strata with stratum-specific prevalence based on the reported prevalence in literature and the geographic proximity of the countries.

The model assumes 99% of the infants infected perinatally were asymptomatic during the acute infection phase, and 90% progressed to chronic hepatitis B, regardless of whether they were symptomatic or not. The model assumed 90% of children infected horizontally before the age of 5 had asymptomatic infection and 70% progressed to chronic hepatitis B. After the age of five, the chance of progressing to chronic hepatitis B was much lower: 70% of infections that occurred after the age of 5 were asymptomatic and only 6% progressed to chronic hepatitis B. Of the acute symptomatic infections, the risks of developing fulminant hepatitis B were 0·1% for perinatal infections, and 0·6% for horizontal infections. The case-fatality rate of fulminant hepatitis was 70% for all ages. Starting from 20 years of age, a small percentage of chronically infected persons (0·5% annually) seroconverted from HBsAg positive to negative, and were no longer at risk of complications related to chronic hepatitis B.

Liver cirrhosis and hepatocellular carcinoma account for the majority of hepatitis B deaths worldwide. The age-specific liver cirrhosis mortality rates were derived from mortality statistics from the United States and Taiwan (China). The age-specific hepatocellular carcinoma incidence was derived by fitting a polynomial function to data from populations with high HBV prevalence, including Alaska Natives, China, the Gambia and Taiwan (China). Given the low survival rates of hepatocellular carcinoma, the death rate of hepatocellular carcinoma was assumed to be the same as the incidence. The rates were adjusted by the prevalence of HBeAg in each country: populations who were HBeAg positive had six times higher the risk of developing hepatocellular carcinoma. The background all-cause mortality rates were from the life table published in the United Nations World Population Prospects.

The lives saved by hepatitis B vaccine were calculated as the difference between predicted deaths of hepatitis B in an unvaccinated cohort and a vaccinated cohort born in a certain year in one country. The vaccination coverages, namely the coverage of the timely birth dose (HepB birth dose within 24 hr of birth) and the coverage of the complete series of at least three doses of hepatitis B vaccine (HepB3) were from the WHO-UNICEF Estimates of National Immunization Coverage (WUENIC) of the past years, and the coverage projection provided by the VIMC secretariat. 95% of infants who received the timely birth dose were assumed to be protected from perinatal infection, and 95% of infants who received the complete series of hepatitis B vaccine (indicated by HepB3 coverage) were assumed to be protected from horizontal infection in their lifetime. The model does not include herd immunity, or the effect of partial vaccination series.

The key uncertainties of the model includes estimates on the prevalence of hepatitis B seromarkers from the pre-vaccination era that are based on limited number of studies in some countries, and the change in mortality of chronic hepatitis B in the long term due to improved access to antiviral treatment. The model used a sensitivity-to-parameters test, rather than a true uncertainty test. It was run with a spread of six parameters that are normally distributed around the original values, with a range of +/- 5%. Two vaccine efficacy parameters (for HepB3 and birth dose respectively) were originally set to 0.95 and then varied together (with the same value for both) between 0.9 and 1.0, normally distributed. These were not country-specific. Four prevalence parameters (HBsAg

prevalence, HBeAg prevalence, anti-HBc prevalence at age 5, anti-HBc prevalence at age 30) all had country group-specific central values, and were varied in unison by +/- 5% around their central values. We assumed the chance of HBV exposure among children who are vaccinated and those who are unvaccinated, and the probability of receiving HepB3 is independent of the birth dose.

## Hib, PCV and Rota – Johns Hopkins University

The Lives Saved Tool (LiST) is a deterministic linear mathematical model for estimating the health impact of changes in health intervention coverage in low- and middle-income countries (LMICs) described in *Walker et al., 2013a*. LiST is a publicly available module within the Spectrum suite, a policy modelling system comprised of several software components. LiST contains over 80 health interventions, including vaccines, and has been used for over a decade to assist in public health decision-making and program evaluation. Evidence-based interventions included in the model have been demonstrated to reduce stillbirths, neonatal deaths, deaths among children aged 1–59 months, maternal mortality or risk factors.

The model describes fixed relationships between inputs (intervention coverage) and outputs (cause-specific mortality or risk factor prevalence) specified in terms of the effectiveness of the intervention for reducing the probability of that outcome under the assumptions that (1) country-specific mortality rates and cause of death structure will not change dynamically, (2) changes in mortality occur in response to changes in intervention coverage, and (3) distal factors, such as improvements in wealth, affect mortality by increasing intervention coverage or reducing risk factors.

The model is built on an underlying demographic projection derived from the United Nations Population Division (UNPD) and age structure for children (0–1, 1–5, 6–11, 12–23, 24–59 months) which serves as a theoretical cohort. Each model uses country-specific inputs of demographic growth (*World Population Prospects, 2019*), under-five mortality rates (*World Population Prospects, 2019*), and cause of death structure (*World Health Organisation, 2021*; *Liu et al., 2016*). Together, these values are used to calculate cause-specific mortality and the potential deaths averted by increasing coverage of interventions. LiST attributes lives saved to changes in coverage of specific interventions, attributing impact first to preventative and then curative interventions, ordered sequentially from periconception, through pregnancy, delivery, followed by the specific age group. By using cause-specific efficacy and applying each intervention to the residual deaths remaining after the previous intervention, LiST ensures that double counting is avoided, and the potential impact of multiple interventions is not erroneously inflated.

Estimates of intervention efficacy are derived from existing reviews, many of which were published in five journal supplements (*Fox et al., 2011*; *Walker et al., 2013a*; *Clermont and Walker, 2017*; *Walker and Friberg, 2017*) National and subnational level impact estimates modelled by LiST have been validated against measured mortality reduction in various LMIC settings and for various packages of interventions (*Amouzou et al., 2010*; *Friberg et al., 2010*; *Hazel et al., 2010*; *Larsen et al., 2011*; *Ricca et al., 2011*; *Victora et al., 2013*).

LiST is used to generate estimates of cases and deaths averted for 0, 1, 2, 3 and 4 years of age due to coverage scale-up of pneumococcal conjugate vaccines (PCV), Hemophilus influenza type b (Hib) vaccine, and rotavirus vaccines. Deaths and cases are calculated separately in LiST. Incidence of diseases (number of cases per child per year) is used instead of cause-specific mortality as a baseline input to calculate cases. Cases and deaths averted by vaccination were calculated by applying estimates of scale-up in coverage in each of the countries. The model accounts for impact of other interventions in each country that could lower the risk of pneumonia, meningitis, or diarrhoea incidence (e.g. clean water and sanitation), or reduce mortality from pneumonia, meningitis, or diarrhoea (e.g. antibiotic treatment) using country-specific coverage of interventions drawing primarily on data from the Demographic and Health Survey (dhsprogram.com) and/or Multiple Indicator Cluster Survey (mics.unicef.org). The specific associations between interventions, risk factors, and mortality within LiST can be accessed via an interactive tool (LiSTVisualizer.org). Deaths and cases averted were calculated holding coverage of all other interventions constant.

The LiST uncertainty bounds are produced using a Monte Carlo approach. For each of the key assumptions in the model we have developed distributions around those values. These include efficacy of interventions, mortality rates, causes of death, relative risks of risk factor for mortality and incidence for severe pneumonia, meningitis and diarrhoea. In general, beta distributions were used

for effectiveness of interventions, correlated normal distribution for mortality rates, Dirichlet distribution for death causes, and log-normal distribution for relative risks. Further information regarding rationale for sampling distributions can be provided upon request.

For the estimates presented here we were asked to only vary efficacy of Hib, PCV, and Rotavirus vaccines, and causes of death of pneumonia, meningitis, and diarrhoea in our uncertainty analysis. For each scenario we were provided 200 sets of varied vaccine efficacies and causes of death from the VIMC Scientific and technical team, based on the 95% confidence intervals of the vaccine efficacies and causes of death. The distribution of model outputs from the 200 runs were then used to produce the uncertainty bounds, which here were set to capture 95% of the distribution of results.

## PCV-specific assumptions

LiST generates estimates of pneumococcal pneumonia and meningitis cases and deaths averted by the coverage scale-up of PCV. The potential envelope of deaths and cases averted by PCV was derived by applying a proxy for the proportion of pneumonia (product of proportion of S. pneumoniae among chest x-ray positive episodes of pneumonia, and proportion of S. pneumoniae due to PCV13 serotypes by region) (*Rudan et al., 2013*; *Johnson et al., 2010*) and meningitis (product of proportion of S. pneumoniae among severe bacterial meningitis cases, and proportion of S. pneumoniae due to PCV13 serotypes by region) (*Johnson et al., 2010*; *Davis et al., 2013*) deaths due to S. pneumoniae in the pre-vaccine era to the country-specific estimates of pneumonia and meningitis mortality. Country-specific incidence of severe pneumonia pre-vaccine introduction was derived from an analysis by Rudan and colleagues (*Rudan et al., 2013*). The country-specific incidence of bacterial meningitis was calculated using the proportions of bacterial meningitis deaths due to S. pneumoniae and Hib from *Davis et al., 2013*, the S. pneumoniae case-fatality rates from *O'Brien et al., 2009*, and Hib case-fatality rates from *Watt et al., 2009*, divided by the total population 1–59 months of age. The proportion of pneumonia and meningitis cases and deaths averted was calculated by applying the 3-dose coverage of PCV scaled by the 80% efficacy of PCV in preventing PCV13 serotypes of invasive pneumococcal disease (*Lucero et al., 2004*), and 84% efficacy of PCV in preventing severe bacterial meningitis (*Davis et al., 2013*), to the fraction of deaths due to S. pneumoniae. The model includes only the direct effect of complete three-dose vaccination coverage.

## Hib vaccine-specific assumptions

LiST generates estimates of Hib pneumonia and meningitis cases and deaths averted by the coverage scale-up of Hib vaccine. The potential envelope of deaths and cases averted by Hib vaccine was derived by applying proxy estimates of proportion of pneumonia (proportion of Hib among chest x-ray positive episodes of pneumonia) and meningitis (proportion of Hib among severe bacterial meningitis cases) deaths due to Hib in the pre-vaccine era to the country-specific estimates of pneumonia and meningitis mortality (*Davis et al., 2013*; *Rudan et al., 2013*). The same country-specific estimates of the incidence of severe pneumonia and meningitis for the PCV impact analysis were used. The proportion of pneumonia and meningitis cases and deaths averted was calculated by applying the three-dose coverage of Hib scaled the 93% efficacy of Hib in preventing invasive pneumococcal disease (*Griffiths et al., 2012*), and the 94% efficacy of Hib in preventing severe bacterial meningitis (*Davis et al., 2013*), to the fraction of deaths due to Hib. The model includes only the direct effect of complete three-dose vaccination coverage.

## Rotavirus vaccine-specific assumptions

LiST generates estimates of rotavirus diarrhoea cases and deaths averted by the coverage scale-up of rotavirus vaccine. The potential envelope of deaths averted by rotavirus vaccine was derived by applying region-specific estimates of the proportion of rotavirus among severe diarrhoea cases and deaths in the pre-vaccine era to the country-specific estimates of diarrhoea mortality (*Walker et al., 2013a*). Region-specific estimates of the incidence of severe diarrhoea were derived from the same source. The proportion of diarrhoea cases and deaths averted was calculated by applying the complete dose coverage of rotavirus vaccine scaled by the region-specific efficacy of rotavirus vaccine in

reducing severe rotavirus gastroenteritis (*Lamberti et al., 2016*) to the fraction of deaths due to rotavirus. The model includes only the direct effect of complete rotavirus vaccination coverage.

## Hib, PCV and Rotavirus – London School of Hygiene and Tropical Medicine (LSHTM)

UNIVAC (universal vaccine decision support model) is a static cohort model with a finely disaggregated age structure (weeks of age <5 years, single years of age 5–99 years). A detailed description of the model and the methods for estimating vaccine impact are available in *Clark et al., 2017*. UNIVAC is available as an R script for desk-based multi-country analyses. It is also available as an Excel-based decision-support model, where it has been widely used by national Ministries of Health in low and middle income countries (LMICs) to estimate the potential impact, cost-effectiveness and benefit-risk of alternative vaccine policy options. In the context of the vaccine impact modelling consortium (VIMC), the R version of UNIVAC was used to generate transparent desk-based estimates of the impact (% reduction in cases, clinic visits, hospitalisations, lifelong sequelae, deaths and DALYs) of three vaccines (haemophilus influenza type b - Hib, pneumococcal and rotavirus) over the period 2000–2030 in 112 LMICs.

Interpolated 1 year time and age estimates (*World Population Prospects, 2019*) were used to calculate the number of life-years between birth and age 5.0 years for each of the 31 births cohorts (2000–2030) in each of the 112 countries. Life-years <5 yrs were multiplied by rates of disease cases and deaths (per 100,000 aged <5 yrs) to estimate numbers of cases and deaths expected to occur without vaccination between birth and age 5.0 years. The rates of disease cases and deaths due to Hib and Pneumococcal were based on estimates generated by *Wahl et al., 2018* for the year 2015. For Hib, these included estimates for non-severe Hib pneumonia, severe Hib pneumonia, Hib meningitis and Hib non-pneumonia/non-meningitis (NPNM) in children aged <5 years. For Pneumococcal, they included estimates for non-severe Pneumococcal pneumonia, severe Pneumococcal pneumonia, Pneumococcal meningitis and severe Pneumococcal non-pneumonia/non-meningitis (NPNM) in children aged <5 years. In addition, the model includes cases of Pneumococcal acute otitis media (AOM) based on estimates by the (*CDC, 2017* and *Monasta et al., 2012*). For both Hib and Pneumococcal, the risk of meningitis sequelae was based on a systematic review and meta-analysis by *Edmond et al., 2010*. For rotavirus, country-specific estimates of rotavirus deaths <5 years were based on the mean of three independent sources of international burden estimates, recently compared in *Clark et al., 2017*. Estimates of rotavirus disease cases (non-severe and severe) were based on systematic reviews and meta-analyses by *Bilcke et al., 2009* and *Walker et al., 2013b*. Granular rotavirus disease age distributions (by week of age <5 years) were based on a recent systematic review and statistical analyses by *Hasso-Agopsowicz et al., 2019*.

Historical time-series estimates of pneumonia and diarrhoea deaths have declined in the absence of vaccination (*Liu et al., 2016*). To avoid over-stating the impact of vaccination, we assume the disease-specific mortality rate will decrease without vaccination at the same rate as the overall under-five mortality rate (*World Population Prospects, 2019*). For consistency, we make the same assumption for Hib/Pneumococcal meningitis and NPNM. We do not assume any decline in the incidence of disease cases, so case fatality ratios (CFRs) decline in each successive year.

Life expectancy estimates by age and year (*World Population Prospects, 2019*) were used to calculate YLLs (years of life lost due to premature mortality) from the age/year of disease death. YLDs (years of life with disease) were calculated by multiplying disability weights by the average duration of illness. DALYs (YLLs + YLDs) were attributed to the year of disease onset.

For all three vaccines, estimates of vaccination impact were restricted to children aged <5 years. The impact was calculated by multiplying the expected number of disease events (cases, clinic visits, hospitalisations, deaths) in each week of age <5 years, by the expected coverage of vaccination in each week of age (adjusted for realistic vaccine delays/timeliness) and the expected efficacy of vaccination in each week of age (adjusted for the waning vaccine protection). The model accounted for partial vaccination by calculating the incremental impact of each dose of vaccination in each week of age. Rotavirus was modelled as a two-dose vaccine co-administered with DTP1 and DTP2 without age restrictions. Hib and Pneumococcal vaccines were modelled as a three-dose vaccine co-administered with DTP1, 2 and 3. For each vaccine, coverage projections by country and year were provided by Gavi, the Vaccine Alliance, over the period 2000–2030 (201910gavi - version 5). Estimates of the timeliness of vaccination (coverage by week of age) were based on the timeliness of DTP1, 2 and 3

reported in USAID Demographic and Health Surveys (DHS) (https://dhsprogram.com) and Multiple Cluster Indicator Surveys (MICS) (mics.unicef.org). Methods for estimating vaccine timeliness have been described previously in *Clark and Sanderson, 2009*. For Hib vaccination, dose-specific efficacy was based on a global systematic review and meta-analysis of RCTs by *Griffiths et al., 2012*. For rotavirus, vaccine efficacy by dose and duration of follow-up (year 1 and year 2) was based on a Bayesian meta-regression of RCTs by *Clark et al., 2019b*. For Pneumococcal, efficacy against all serotypes of pneumococcal disease (vaccine type and non-vaccine type) was based on a global meta-analysis by *Lucero et al., 2004*.

UNIVAC is not a transmission dynamic model, and thus excludes indirect effects (both positive and negative). This is likely to lead to substantial under-estimates of impact in some countries, particularly for Hib vaccine. More detailed validation against real-world post-introduction evidence of impact is needed. However, the available data in many of the countries included in this desk-based analysis are insufficient to allow validation of modelled estimates (against real-world estimates of post-introduction vaccine impact) and/or parameterisation of a country-specific transmission dynamic model. As such, there is a good deal of uncertainty in the predicted estimates for many countries.

## Human Papilloma Virus (HPV) – Harvard University

The Center for Health Decision Science companion model is a flexible tool that has been developed to reflect the main features of HPV vaccines, and to project the potential (health and economic) impacts of HPV vaccination at the population level in settings where data are very limited (*Goldie et al., 2008*). The model is constructed as a static cohort simulation model based on a structure similar to a simple decision tree and is programmed using R software (*R Development Core Team, 2020*). The model tracks a cohort of girls at a target age (e.g., 9 years) through their lifetimes, comparing health and cost outcomes with and without HPV vaccination programs. Population-level analyses are conducted by running multiple cohorts.

Unlike more complex empirically-calibrated micro-simulation models (*Goldie et al., 2007*; *Campos et al., 2012*; *Kim et al., 2007*), the companion model does not fully simulate the natural history of HPV infection and cervical carcinogenesis. Instead, based on simplifying assumptions (i.e., duration and stage distribution of, and mortality from, cervical cancer), which rely on insights from analyses performed with the micro-simulation model, and using the best available data on setting-specific age-specific incidence of cervical cancer and HPV-16/18 type distribution and assumed vaccine efficacy and coverage, the model estimates reductions in cervical cancer risk at different ages. By applying this reduction to country-specific, age-structured population projections incorporating background mortality (*World Population Prospects, 2019*), the model calculates averted cervical cancer cases and deaths, and transforms them into aggregated population health outcomes, years of life saved and disability-adjusted life years (DALYs) averted. DALYs are calculated using the approach adopted by the Global Burden of Disease (GBD) study (*Murray and Lopez, 1996*), using stage-specific disability weights. The model also incorporates five-year stage-specific survival probabilities for untreated and treated cervical cancers (by region) and treatment; access proportions (by country). These values are combined into weighted averages to provide country-specific 5 year survival parameters, matched to GLOBOCAN 2020 age-specific mortality rates .

The companion model captures the burden of HPV infection by estimating the number of cervical cancer cases caused by HPV infection based on epidemiological data obtained from various sources (*Goldie et al., 2008*). The model assumes that age-specific cervical cancer incidence, average age of sexual initiation, and the level of other risk factors remain constant over the time horizon of the model. It assumes that girls are fully immunized and that girls effectively immunized against vaccine-targeted HPV types can develop cervical cancer associated with non-vaccine HPV types; also, no cross-protection against non-vaccine types is assumed. Country-specific assumptions are used for the proportion of cancer that is attributed to the vaccine-covered types (HPV-16/18 in this analysis) (*Guan et al., 2012*). Vaccine-induced immunity is assumed to be lifelong. Currently, there are no interactions or correlations between doses as the model assumes fully vaccinated individuals (whether with 1, 2, or three doses). All assumptions are varied in sensitivity analyses.

Four key parameters were identified for probabilistic sensitivity analysis (PSA): HPV-16/18 type distribution, age-specific cervical cancer incidence, stage distribution of cervical cancer, and stage-specific 5 year survival and treatment access (as a combined parameter). Each parameter was

assigned a $\beta$-PERT distribution for probabilistic sampling, with the bounds determined by: (1) empirical data for type distribution; (2) confidence intervals estimated from cervical cancer cases in GLOBOCAN 2020 (*Ferlay et al., 2020*) (3) assumed +/- 10% bounds from the base case for stage distribution; and (4) assumed +/- 10% bounds from the base case for stage-specific probability of death following 5 year survival, if this estimate is contained between zero and one. For stage distribution, for a single parameter set, the value for a single stage (specifically, stage 2, given country-level differences in survival and disability weights) is drawn individually and the remaining stages are normalized to adjusted values in order for the four stages to add up to one. As the stage-specific probability of death cannot exceed 100%, the minimum bound is set to the difference between the base-case value and one, where relevant. Two hundred independent parameter sets were drawn for each country.

## Human Papilloma Virus (HPV) – London School of Hygiene and Tropical Medicine (LSHTM)

The Papillomavirus Rapid Interface for Modelling and Economics (PRIME) is a static, proportional impact model that can estimate the impact of HPV vaccination on cervical cancer cases, deaths, and disability-adjusted life years as well as the cost-effectiveness of vaccination programmes at the global, regional, and national levels (*Abbas et al., 2020a*; *Jit et al., 2014*). The PRIME model was developed by LSHTM in collaboration with the World Health Organization (WHO), Laval University and Johns Hopkins University. It is designed to estimate the impact and cost-effectiveness of HPV vaccination in low- and middle-income countries (LMICs). In addition to its application in the Vaccine Impact Modelling Consortium (VIMC), it has been used to support vaccine recommendations by WHO, as well as individual countries. It has been validated against published studies using HPV vaccine economic models set in LMICs (*Jit et al., 2014*). It was also endorsed by the WHO's expert advisory committee, the Immunization and Vaccines Implementation Research Advisory Committee (IVIR-AC) to provide a conservative estimate of the cost effectiveness of vaccinating girls prior to sexual debut.

The Excel-based version of the model and documentation are publicly accessible at http://prime-tool.org/ for use by country programme managers and planners to facilitate country-specific decision-making in LMICs. The R package of the model (prime) has additional functionality such as multiple cohorts and probabilistic sensitivity analysis and is available at https://github.com/lshtm-vimc/prime (copy archived at swh:1:rev:0da13630968de0863f38294a1c234c5947baf97e); *Abbas and Hadley, 2021*. It can be used for research, global analyses and to generate the vaccine impact estimates used by VIMC. Data inputs include country and age-specific cervical cancer incidence, prevalence, and mortality among females. The model estimates vaccination impact in terms of reduction in age-dependent incidence of cervical cancer and mortality in direct proportion to vaccine efficacy against HPV 16/18, vaccine coverage, and HPV type distribution. It assumes that vaccinating girls prior to infection with HPV types 16 and 18 fully protects them from developing cervical cancer caused by HPV 16 and 18, in accordance with vaccine trials (*Schiller et al., 2012*).

The model assumes a two-dose schedule with perfect timeliness at the target ages given in the coverage estimates. Herd effects are not considered meaning that the vaccine impact estimates produced are conservative, although the model can be used in conjunction with transmission dynamic models to project indirect effects. The impact of vaccinating multiple age cohorts is estimated by using the most conservative assumption that 9–14 year old girls who have sexually debuted are not protected, although these assumptions do not change the overall impact estimates significantly (*Jit and Brisson, 2018*).

## Measles – London School of Hygiene and Tropical Medicine (LSHTM)

DynaMICE (DYNAmic Measles Immunisation Calculation Engine) is a measles transmission and vaccination model developed by LSHTM with input from Harvard University and the University of Montreal (*Verguet et al., 2015*). It has been previously used to inform policies on measles-containing vaccines by WHO; this work has been reviewed by the WHO's Immunization and Vaccines Implementation Research Advisory Committee (IVIR-AC) as well as WHO's Strategic Advisory Group of Experts on Immunization (SAGE)'s measles and rubella working group.

It is an age-structured compartmental transmission dynamic model with compartments for maternal immune, susceptible, infected, recovered, and vaccinated subpopulations. A proportion of infected people will die depending on their age and country characteristics (*Portnoy et al., 2019*). The population is also stratified by age with weekly age classes up to age 3 years, and annual age classes thereafter up to 100 years. The force of infection is calculated by combining an age-dependent social contact matrix from the POLYMOD study (*Mossong et al., 2008*), demographic distribution for each country, and an estimated probability of transmission per contact. The probability of transmission per contact is then estimated from the basic reproduction number of measles using the principal eigenvalue method. Vaccination is incorporated as a pulse function and can be delivered to any age or range of ages and in either routine or through supplementary immunisation activities (SIAs) or campaigns. The ability of SIAs to reach children who miss routine vaccination is determined using analyses of Demographic and Health Survey (DHS) data (*Portnoy et al., 2018*). Vaccine efficacy is dependent on age and the number of doses received (*Hughes et al., 2020*). The model has been previously described in detail (*Verguet et al., 2015*).

## Measles – Pennsylvania State University (PSU)

The PSU measles model is a dynamic, age-structured, discrete time-step, annual SIR model. Unlike conventional SIR models, which describe dynamics at the scale of an infectious generation (*Finkenstadt and Grenfell, 2000*) or finer (*Anderson and May, 1991*), it models the aggregate number of cases over one-year time steps. While this is coarse relative to the time scale of measles transmission, it matches the annual reporting of measles cases available for all countries, since approximately 1980, for all countries through the WHO Joint Reporting Form (JRF). To account for the fine-scale dynamics that are being summed over a full year, the model describes the number of infections ($I_{i,t}$) in country $i$ and year $t$, and age class $a$ as an increasing function of the fraction, $p_{i,t}$, of the population susceptible in age class $a$ at the start of year $t$, $S_{i,t}$:

$$E[I_{a,i,t}] = p_{i,t} * S_{a,i,t},$$

where $E[\cdot]$ indicates the expectation and $p_{i,t}$ is a country and year specific annualized attack rate modeled as:

$$p_{i,t} = invlogit\left(-\beta_{0,i} + \beta_{1,i} * \frac{\sum_a S_{i,t}}{N_{i,t}} + e_t\right),$$

where invlogit() indicates the inverse logit function, $N_{i,t}$ is the total population size in country $i$ and year $t$ over all age classes, and $e_t$ is a Gaussian random variable with mean 0 and variance $\sigma^2$. The parameters $\beta_{0,i}$, $\beta_{1,i}$, and $\sigma^2$ are fit to each country independently using a state-space model fitted to observed annual cases reported through the JRF from 1980 to 2016 as described by *Eilertson et al., 2019*. Inferred cases are not constrained by external mortality estimates. Historical population and vaccination coverage values are provided by WHO as described by *Simons et al., 2012*.

The number of susceptible individuals in each single-year age class $a$ ($a$=2,..., 100) is equal to the number not infected in the previous year, nor immunized through supplemental immunization activities (SIAs). The number susceptible is further deprecated by the crude death rate. The efficacy of doses administered through SIAs is assumed to be 99%; SIA doses are assumed to be independent of prior routine immunization. The number of susceptible individuals in age class $a$=one is assumed to be 50% of the annual live birth cohort; this assumes that all children have protective maternal immunity until 6 months of age. Age class $a$=two and $a$=m is assumed to receive a first and second dose (respectively) of routine measles vaccination before the start of the time step. We assume that the second routine dose is delivered only to those who have received the first routine dose. Efficacy is assumed to be 85% and 93% for the first dose in countries delivering at 9 m and 12 m of age, respectively, and assumed to be 99% for the second dose.

Deaths are calculated by applying an age- and country-specific case fatality ratio (CFR) to each country. CFRs for cases below 59 months of age for all countries were taken from *Portnoy et al., 2019*; CFR for cases above 59 months of age are assumed to be 50% lower than those applying to under-5s.

Forward simulations of this model assume random variation in the annual attack rate according to the parameter $\sigma^2$. Future vaccination coverage values, for routine and SIAs, are assumed known and future birth and death rates are assumed known.

## MenA - University of Cambridge

The University of Cambridge MenA model is a compartmental transmission dynamic model of Neisseria meningitidis group A (NmA) carriage and disease to investigate the impact of immunisation with a group A meningococcal conjugate vaccine, known as MenAfriVac, as published by *Karachaliou et al., 2015*. The model is age-structured (1 year age groups up to age 100) with continuous ageing between groups. Model parameters were based on the available literature and African data wherever possible, with the model calibrated on an ad-hoc basis as described below.

The population is divided into four states, which represent their status with respect to the meningitis infection. Individuals may be susceptible, carriers, ill or recovered, and in each of these states be vaccinated or unvaccinated, with vaccinated individuals having lower risks of infection (carriage acquisition) and disease (rate of invasion). We assume that both carriers and ill individuals are infectious and can transmit the bacteria to susceptible individuals. The model captures the key features of meningococcal epidemiology, including seasonality, which is implemented by forcing the transmission rate, the extent of which varies stochastically every year.

Since only a small proportion of infected individuals develop the invasive disease, disease-induced deaths are not included in the model. From each compartment, there is a natural death rate from all causes. Carriage prevalence and disease incidence vary with age, and the model parameterised these distributions using a dataset from *Campagne et al., 1999*; the case:carrier ratio consequently varies with age. The duration of 'natural immunity' is an important driver of disease dynamics in the absence of vaccination but good data on this parameter is lacking; instead, prior estimates are used (*Irving et al., 2012*).

The model assumes that mass vaccination campaigns occur as discrete events whereas routine immunisation takes place continuously. We allowed the duration of protection to vary uniformly between 5 years and 20 years for the 0–4 year-olds and 10–20 years for over 5-year-olds. For the 200 runs, we selected pairs of values for these two parameters so that duration of protection for the older age group is not shorter than the duration of protection for 0–4 year-olds (*Yaro et al., 2019*; *White et al., 2019*). Vaccine efficacy against carriage and disease is 90%.

Disease surveillance is not comprehensive across the meningitis belt, so the disease burden is uncertain in several countries. Therefore, the model classifies the countries into three categories, based on the incidence levels using historical data. This classification defines the transmission dynamic parameters. The model generates estimates of case incidence, to which a 10% case-fatality ratio is applied to estimate mortality (*Lingani et al., 2015*). To estimate DALYs it is assumed that 7·2% of survivors have major disabling sequelae with a disability weight of 0·26 (*Edmond et al., 2010*).

Countries were stratified into high and medium risk, and different infection risks applied based on this stratification. As there was insufficient information to define infection risk on a country-by-country basis, the approach/stratification was agreed upon with experts in the WHO meningitis team. For countries only partly within the meningitis belt, only the (subnational) area at risk was included.

To produce estimates on the impact of vaccination, 200 simulation runs were generated by stochastically varying the baseline transmission rate to reflect between-year climactic or another external variability. Although each individual simulation reflects the reality of irregular and periodic epidemics, as visually compared to time series from Chad and Burkina Faso and analysis of inter-epidemic periods, the resulting averaged estimates give a stable expected burden of disease over time. Uncertainty in other model parameters is currently not quantified.

## MenA – Kaiser Permanente Washington

This model is a stochastic, age-structured, compartmental model of the transmission of serogroup A Neisseria meningitidis (MenA) (*Jackson et al., 2018*). Model compartments track hosts' status with respect to MenA exposure (as Susceptible, Colonized, and Invasive Disease) and adaptive immunity

to infection/disease (as High, Low, or No immunity). Exposure to MenA through colonization leads to the 'low immunity' state, in which individuals are still susceptible to colonization but have a reduced risk of developing invasive disease if colonized. MenA colonization among individuals with low immunity leads to a 'high immunity' state, which is highly protective against both colonization and disease.

Model parameters such as the age-specific force of infection, rates of immune waning, and immune protection against colonization/disease were estimated by approximate (*Marjoram et al., 2003*). Prior distributions for model parameters were taken from the existing literature (*Marjoram et al., 2003*). Simulated prevalence of colonization by age was compared to longitudinal studies of the prevalence of MenA colonization in Burkina Faso (*Traore et al., 2009*) and the expected age distribution of MenA cases (*Campagne et al., 1999*).

In the simulations, mortality burden estimates are obtained in a 'bottom-up' manner, in that case fatality ratios (CFRs) (*Campagne et al., 1999*; *Boisier et al., 2007*; *Belcher et al., 1977*; *Traore et al., 2009*; *Varaine et al., 1997*; *World Health Organisation, 2001*; *World Health Organization, Geneva, 2005*) are applied to simulated case counts.

For estimating the impact of serogroup A polysaccharide conjugate vaccine (MenAfriVac), vaccination is assumed to be superior to natural immunity, based on estimates of vaccine effectiveness and serum bactericidal antibody (SBA) concentrations (*Goldschneider et al., 1969*), an assumption with is shown to better capture the dynamics of NmA in Burkina Faso following mass vaccination campaigns with MenAfriVac, compared to assuming vaccination is equivalent to natural infection (*Jackson et al., 2018*).

Countries were stratified based on risk (hyper-endemic vs. not), and different forces of infection used based on risk group. For countries only partly within the meningitis belt, the model was restricted to the area at risk.

Variability in infections rates is represented by randomly sampling values for the force of infection parameters within Â±20% of their estimated values; new values are sampled annually to reflect annual variation in climate or other external factors. To estimate incidence of MenA disease and death in each modelled vaccination scenario, we run 200 iterations of the simulation. Each simulation randomly samples parameter values from their posterior distributions; mean estimates are taken from the mean values by year and age across the 200 iterations. Key sources of uncertainty not presently included are the expected duration of vaccine-induced protection and the force of infection in countries for which MenA surveillance data are lacking.

## Japanese Encephalitis – National University of Singapore

This deterministic dynamic model uses a basic catalytic model for the force of infection (FOI), in which individuals become infected and are then immune. Vaccination is modelled as a removal of susceptibles from the susceptible class. As humans are dead-end hosts for Japanese encephalitis (JE), infection comes from animal reservoirs via mosquitoes. This simple model successfully captures the natural history and transmission dynamics of JE.

A systematic review of all published studies and publicly available JE surveillance data was undertaken to collate a dataset of age-stratified case data. The FOI model is fit to age-stratified national surveillance data that were publicly available and data identified via this systematic review of age-stratified JE case data (*Quan et al., 2020*). Data from a total of 10 countries and 17 studies was used, which gave estimates of a wide range of force of infection parameters, as expected for the wide geographical locations. The model is fit in a Bayesian framework using RStan. This gave FOI estimates for all locations in which data is available. For areas in which data was not available we extrapolated from areas in which it was, using the WHO groupings of transmission intensity (*Campbell et al., 2011*). In order to generate uncertainty in the case burden estimates, all model parameters were sampled from the posterior distributions of the parameter estimates. The symptomatic rate was sampled from uniform distribution (0·002, 0·004) SAGE Working Group on Japanese encephalitis (*World Health Organization, 2014*), the proportion of these symptomatic cases that died was from uniform distribution (0·2, 0·3) and the force of infection from the relevant posterior estimate from the age-stratified case data described above. The vaccine was assumed to be 100% effective, and protection lifelong.

Disease burden was generated from the 'bottom up': i.e. from infection rates applying parameters governing the proportion of infections that are symptomatic and the proportion that die (case fatality ratio). The key uncertainties which affect disease burden estimates are the method of extrapolation of FOI from areas in which there is data to areas in which there is not. Spatial modelling work is on-going to improve this extrapolation and to make estimates on smaller spatial scales. The case fatality ratio is also uncertain, and further work undertaking a systematic review of this is ongoing.

## Japanese Encephalitis - University of Notre Dame

We developed a static, stochastic model of Japanese encephalitis virus (JEV) transmission with a constant force of infection (FOI) to estimate the burden of JE and the potential impact of vaccination in JE-endemic countries. JEV is a mosquito-transmitted, zoonotic pathogen that requires an animal host for ongoing transmission, given humans are believed to be a dead-end host (*van den Hurk et al., 2009*). Therefore, JE incidence is limited to geographic regions where there are suitable hosts and vectors to sustain both ongoing transmission in animal hosts and spill-over to humans. To estimate the number of JEV infections, the model first estimates the number of people at risk of infection, and then estimates the transmission intensity in each country. JE burden (including cases, deaths, and DALYs) was then estimated from the number of JEV infections. Key sources of uncertainty in our model are the spatial variation in JEV transmission intensity and the proportion of JEV infections that result in either a severe case or death.

To identify the areas suitable for sustained JEV transmission, and the size of the population living in at-risk areas, a spatial analysis of the risk factors associated with JEV was conducted. Potential JEV-endemic areas were identified using large-scale spatiotemporal datasets related to suitable climate conditions for the vector species, suitable habitat conditions for the vector, and the presence of potential zoonotic hosts. Transmission was assumed to occur only in areas occupied by the primary vector, *Culex tritaeniorhynchus* (*Longbottom et al., 2017*), or where the annual minimum temperature exceeded 20°C and annual precipitation exceeded 150 cm. Suitable habitat conditions included areas with rice cultivation or nearby wetlands (*Gumma, 2011*). Within these suitable areas, people were considered at risk of infection if the density of domestic pigs or fowl exceeded two per km (with uncertainty represented by varying the animal threshold from 0 to 10 per km) (*Robinson et al., 2014*). Risk maps were validated using seroprevalence and surveillance data.

Next, the FOI in each country was estimated from age-specific incidence data using a catalytic model. FOI represents the per-capita rate at which susceptible individuals are infected. Age-specific incidence data was obtained from a literature search, restricted to studies conducted in areas with no history of vaccination (or prior to documented vaccination) to simplify the estimation process. For several countries where no age-specific incidence data was available, FOI estimates were drawn from the posterior estimate of a neighbouring country. FOI estimates for each study were estimated using a maximum likelihood approach, using the observed numbers of JE cases per age class in each year. Study-specific FOI values were estimated using a Bayesian framework via a Markov chain Monte Carlo (MCMC) approach implemented in the software package STAN. The probability of asymptomatic infection and the case fatality ratio for symptomatic infections were assumed to be independent of the FOI.

The annual number of JEV infections for a given study area were then calculated from the FOI estimate and the size of the at-risk population. In the absence of vaccination, the number of infections in age class was calculated by multiplying the age-specific probability of infection by the number of at-risk individuals in the age class. Vaccination reduced the number of at-risk individuals in each targeted age class based on provided coverage estimates. We assume that all vaccinated individuals receive a full vaccine regimen and that routine and campaign-based vaccinations are independent. The number of JE cases and deaths were then estimated from the number of JEV infections based on the proportion of infections that are symptomatic or fatal. The number of JE cases was modelled assuming an asymptomatic to symptomatic (A:S) ratio for JEV infections of 295:1 (95% CI: 83:1 to 717:1), based on the range from published estimates (*van den Hurk et al., 2009*). The case fatality ratio (CFR) from JE was assumed to follow a Beta distribution with a median symptomatic CFR of 0·33 (95% CI: 0·05-0·75), reflecting the large uncertainty in this parameter (*van den Hurk et al., 2009*). The annual burden of JE at the national-level was calculated using

disability-adjusted life years (DALYs) with disability weights taken from the Global Burden of Disease 2016 report (*GBD 2016 DALYs and HALE Collaborators, 2017*).

## Rotavirus – Emory University

The Emory model is a dynamic, deterministic, age-structured compartmental transmission model simulates rotavirus transmission and estimates disease incidence/burden in a given country. The model is based on a Susceptible–Infected–Recovered (SIR) structure, with elaborations in order to capture the complexities of rotavirus immunity and transmission. In particular, individuals can be infected up to four times. We model the following age groups: 0–1 months, 2–3 months, 4–11 months, 1 year age bands from 1 to 4 years old, and 5 years and older. We use realistic, age-specific population sizes, aging and death rates.

In the model, infants are born with maternal immunity (*Linhares et al., 1989*). After maternal immunity wanes, infants become susceptible to a primary rotavirus infection. We assume that protection is conferred by previous infections against subsequent infections, such that the proportion of individuals that remain susceptible to re-infection decreases with each subsequent infection (*Gladstone et al., 2011*; *Velázquez et al., 1996*). Primary, secondary, tertiary, and quaternary infections are assumed to have the same duration of infectiousness, however non-primary infections have lower per-contact infectiousness relative to primary infections (*Pitzer et al., 2012*). Immunity is assumed to be a mix of 'take type'where a portion of individuals develop long-term immunity, while others remain fully susceptible to subsequent infections. We assume primary, secondary, tertiary, and quaternary infections had different probabilities for developing rotavirus gastroenteritis (*Velázquez et al., 1996*). We assume only severe rotavirus gastroenteritis cases are reported to surveillance and can result in death. This model incorporates the introduction of vaccines in a specified year, delivered to 2- and 4 month olds. We assume that vaccine doses are independent. We incorporate an immunogenicity parameter that determines whether individuals will respond to the vaccine (*Patel et al., 2009*). If individuals respond to vaccination, we assume that the vaccine acts like a natural infection; the probability of becoming infected, given vaccination and natural history of infections, goes down with each subsequent vaccine dose and natural infection. Values for natural history parameters were set to values identified in birth cohort and challenge studies.

In lieu of fitting this model to estimate country-specific effective contact rates, we used a linear regression model to estimate the mean age of severe rotavirus infection, and subsequently calculated the basic reproduction number. Variables considered for inclusion were: under five mortality rates (*World Bank, 2021*), gross domestic product (GDP) per capita (*World Bank, 2021*), total GDP (*World Bank, 2021*), region, sub-region, birthrate (*World Population Prospects, 2019*), life expectancy (*World Population Prospects, 2019*), and percent of the population living in a rural setting (*World Bank, 2021*). The linear regression model was fit using a training (80%) and validation (20%) data set, with the optimal model for each country being selected to optimize the correlation accuracy and the mean absolute percent error (MAPE). The basic reproduction number was calculated by dividing the life expectancy for each country (*World Bank, 2021*) by the fitted average age of severe infection.

To account for uncertainty, we generated 200 parameter sets by uniformly sampling from the published range of vaccine immunogenicity and the 95% confidence interval of the regression models estimated mean age of infection for each country. The remaining parameters were fixed. We then simulated the model for each set of fixed and sampled parameters. We calculated the central burden/impact estimate as the median of the 200 probabilistic runs. To calculate the number of deaths, we estimated the number of previous rotavirus infections (with first infections being most severe and subsequent infections being less likely to cause severe disease) and then multiplied this quantity by the estimated rotavirus case fatality ratio for each country and age group, based on data from the Global Burden of Disease Study (*Troeger et al., 2018*).

## Rubella – Johns Hopkins University

We developed a discrete-time stochastic age-structured compartmental rubella transmission model, building from previous work describing rubella dynamics (*Metcalf et al., 2012a*). The key feature of the model is a matrix that at every time-step defines transitions from every combination of

epidemiological stage (maternally immune 'M', susceptible 'S', infected 'I', recovered 'R', and vaccinated 'V', taken to indicate the effectively vaccinated) and age group (1 month age groups up to 20 years old, then 1 year age groups up to 100 years old) to every other possible combination of epidemiological stage and age group. The discrete time-step was set to about two weeks (i.e. 24 time steps in a year), the approximate generation time of rubella.

Demographic parameters (population size, crude birth rates, and age-specific death rates) and vaccination coverage were time and country-specific, and were supplied by VIMC. We assumed dependence between routine vaccine doses. We adjusted campaign coverage based on the assumptions that a portion of the population may always remain inaccessible to campaigns. We assumed the age and time-specific proportion inaccessible corresponds to WUENIC DTP routine vaccination rates World Health Organization and UNICEF (*World Health Organization and UNICEF, 2015*) if it does not exceed routine coverage. Duration of maternal immunity (*Nicoara et al., 1999*) and vaccine efficacy (*Boulianne et al., 1995*) were assumed from published literature and are constant across time and country. The annual introduction of infected individuals was scaled with the median time-specific population size of each country set to trigger an outbreak if the size of the susceptible population was large enough but small enough to not effect probability of elimination.

Country-specific transmission to individuals in age group $a$ from individuals in age group $j$ for each time-step $t$ is defined by $\beta_{a,j,t} = \overline{\beta_{a,j,}}(1 + \alpha cos(2\pi t))$, where $\overline{\beta_{a,j}}$ is mean transmission from individuals in age group $j$ to age group $a$, and $\alpha$ is a parameter controlling the magnitude of seasonal fluctuations (assumed 0.15 [*Metcalf et al., 2012b*] and constant over time and country). Mean transmission from individuals in age class $j$ to age class $a$, $\overline{\beta_{a,j}}$, was estimated by rescaling population-adjusted age-contact rates (time constant and country-specific, *Prem et al., 2017*) to reflect the assumed basic reproductive number ($R_0$) of rubella. $R_0$ distributions were country-specific and estimated by fitting a dampened exponential model (*Farrington, 1990*) with likelihood-based MCMC to published rubella immunoglobulin G (IgG) seroprevalence data.

Rubella burden is generated from a 'bottom up'approach in which we calculate CRS cases, and deaths, from modeled output. Country- age- and time-specific CRS cases were estimated by multiplying the country, age and time-specific number of susceptible individuals, the country and time-specific sex ratio of the population, the country- and age-specific fertility rate, the country- age- and time-specific probability of becoming infected over 16 week period, and finally the probability of CRS following rubella infection during the first 16 weeks of pregnancy (estimated ~0.4; *Andrade et al., 2006*; *Hahné et al., 2009*; *Grillner et al., 1983*; *Miller, 1991*; *Mirambo et al., 2019*; *Zgórniak-Nowosielska et al., 1996*; *Vejtorp and Mansa, 1980*). Fetal deaths were estimated directly from rubella infections among women in the first 16 weeks of pregnancy as 20.7 per 100 (*Mirambo et al., 2019*; *Cooper and Krugman, 1967*; *Miller et al., 1982*; *Siegel et al., 1966a*; *Siegel et al., 1966b*) and infant deaths were estimated from the number of CRS cases as 8.9 per 100 (*Miller, 1991*; *Cooper and Krugman, 1967*; *Saad de Owens and Tristan de Espino, 1989*; *Panagiotopoulos et al., 1999*; *Toizumi et al., 2014*).

We simulated 200 stochastic runs for each country from the year 1980 to 2100. Model uncertainty includes process uncertainty for all epidemiological and demographic transition and uncertainty from the $R_0$, CRS rate, and CRS death rate distributions. Model input parameters (e.g., $R_0$) were fit to empirical data, however the mechanistic transmission model itself is not directly fit to data.

## Rubella – Public Health England

This is an age and sex-structured, deterministic, compartmental model of the transmission dynamics of rubella (*Vynnycky et al., 2016b*; *Vynnycky et al., 2019*). The population is stratified into those with maternal immunity (lasting 6 months), susceptible, pre-infectious (infected but not yet infectious), infectious and immune, using annual age bands and a 'Realistic Age Structure' (*Schenzle, 1984*). Country-specific birth and age-specific death rates were fixed at 2010 levels and calculated from UN population survival data for 2010–15 (*UNWPP, 2017*) respectively. The supplement to *Vynnycky et al., 2016a* provides the model's differential equations.

The force of infection (rate at which susceptibles are infected) changes over time and is calculated using the number of infectious individuals and the effective contact rate (rate at which infectious and susceptible individuals come into effective contact). Contact is described using the following matrix of 'Who Acquires Infection From Whom':

$$\begin{pmatrix} \beta_1 & 0.7\beta_2 \\ 0.7\beta_2 & \beta_2 \end{pmatrix}$$

The effective contact rate differs between <13 and ≥13 year olds, with its relative size based on contact survey data (*Mossong et al., 2008*). $\beta_1$ and $\beta_2$ are calculated from the average force of infection in <13 and ≥13 year olds, estimated from age-stratified rubella seroprevalence data, which had been collected before rubella containing vaccine (RCV) was introduced (*Vynnycky et al., 2016b*). Seroprevalence data were available for 28 countries (see *Vynnycky et al., 2019*). For countries lacking seroprevalence data, we used data from countries in the same WHO region (*Vynnycky et al., 2016b*; *Vynnycky et al., 2019*). Confidence intervals (CI) on the force of infection were calculated using 1000 bootstrap-derived-seroprevalence datasets (*Vynnycky et al., 2016b*; *Vynnycky et al., 2019*). The vaccine doses were assumed to be correlated, with 100% of those vaccinated previously being vaccinated I SIAs, where possible and 50% of those who have received RCV1 receiving RCV2, where possible.

Country-specific numbers of congenital rubella syndrome (CRS) cases in year $y$ during 2001–2080 were calculated by summing the number of CRS cases born each day to women aged 15–49 years. As assumed elsewhere (*Vynnycky et al., 2003*; *Vynnycky et al., 2016b*; *Vynnycky et al., 2019*), infection during the first 16 weeks of pregnancy carries a 65% risk of the newborn having CRS. The number of CRS deaths in year $y$ was calculated by multiplying the number of CRS cases born in year $y$ by the assumed case fatality rate (30%). The latter was assumed to have a plausible range of 10–50%, consistent with the number of DALYs for cases in year $y$ was calculated by multiplying the number of CRS cases in year $y$ by the corresponding DALY (*Simons et al., 2016*), which was based on the country-specific World Bank Income group for 2017 (*World Bank, 2017*). Both the DALYs and the assigned World Bank income group remained fixed over time. As rubella infections are mild, rubella-specific deaths are not included and people with rubella infection are assumed to die at the general all-cause, age and sex-specific mortality rate.

Confidence intervals on the outputs for each setting were calculated as the 95% range of the outputs obtained by running the model using 200 combinations of 5 randomly-sampled parameters. The parameters were the pre-vaccination force of infection which was used to calculate the contact parameters (see above), the risk of a child being born with CRS if his/her mother had been infected during pregnancy, the CRS-related case-fatality rate, the vaccine coverage and the vaccine efficacy. The pre-vaccination force of infection was sampled from 1000 bootstrap-derived force of infection estimates, obtained by fitting catalytic models to bootstrap-derived seroprevalence data for that setting, or, if that setting lacked seroprevalence data, from bootstrap-derived force of infection estimates from countries in the same WHO region as the country of interest (*Vynnycky et al., 2016b*; *Vynnycky et al., 2019*). The remaining parameters were randomly sampled from distributions reflecting their plausible range, as implied by published studies, wherever possible (*Vynnycky et al., 2016a*). For example, the CRS-related mortality was sampled from the uniform distribution in the range 10–50%, consistent with estimates from three studies in Vietnam, Greece and Panama, in which the 95% confidence intervals were 20–51%, 12–50% and 15–40% respectively (*Toizumi et al., 2014*; *Panagiotopoulos and Georgakopoulou, 2004*; *Saad de Owens and Tristan de Espino, 1989*). The risk of a child being born with CRS to a mother infected in the first 16 weeks of pregnancy was sampled from the Gamma distribution with shape and scale parameters 37 and 56 respectively. This assumption leads to a median and 95% range of 65% and 47–88% respectively for this risk, consistent with estimates from several studies (*Miller et al., 1982*; *Grillner et al., 1983*; *Hahné et al., 2009*) which, as found in a recent review (*Thompson et al., 2016*) were likely to have been more reliable than those in other studies. The sampling was conducted assuming that the parameters were independent.

## Yellow Fever – Imperial College London

The Imperial College yellow fever (YF) transmission model is a static force of infection (FOI) epidemiological model. The first iteration was originally published by *Garske et al., 2014*; however, this has been extensively updated by *Gaythorpe et al., 2021a* to provide the 2019 model estimates. The model is fitted at the first administrative level or province level for all countries considered at risk or endemic for YF. In each administrative unit, the force of infection is assumed to be constant across

the observation period and across age groups. This is analogous to assuming that all yellow fever transmission occurs as a result of spillover events from the sylvatic reservoir. As a result, this model variant includes no herd immunity effects.

The model is estimated from multiple data sources which inform separate components. A generalised linear model, based on environmental covariates, is informed by presence/absence of yellow fever reports between 1984 and 2019 at province level. Reports of yellow fever are based on outbreak reports published by the WHO and on cases reported in the Yellow Fever Surveillance Database (YFSD) managed by WHO-AFRO, to which 21 countries in West and Central Africa contribute, and reports from the Brazilian Ministry of Health as well as PAHO (*World Health Organization, 2021a*, *World Health Organization, 2021b*). The environmental covariates were revisited since the initial 2014 model version and now include substantially updated datasets. Covariates include non-human primate species occurrence, vector occurrence, temperature, land cover type and altitude (*Fick and Hijmans, 2017*; *Kraemer et al., 2015*; *LP DAAC NASA, 2001*; *Xie and Arkin, 1996*; *IUCN, 2019*). The regression model provides estimates of the probability of yellow fever reports across the endemic zone. In order to preserve structural uncertainty from covariate selection, we average over the 20 best fitting generalized linear models within the Bayesian framework.

These estimates are then translated to the number of infections by further fitting to data obtained from 42 serological surveys performed in Africa (*Chepkorir et al., 2019*; *Diallo et al., 2010*; *Kuniholm et al., 2006*; *Merlin et al., 1986*; *Omilabu et al., 1990*; *Tsai et al., 1987*; *Werner et al., 1985*). In each survey location, a static, age-independent force of infection is fitted. This is also informed by estimates of demography and vaccination coverage including historic vaccination campaigns (*World Health Organization and UNICEF, 2015*; *Hamlet et al., 2019*). We assume there is no correlation between vaccine doses.

Model components are estimated within a Bayesian framework with adaptive Markov Chain Monte Carlo sampling, this framework was extended in Gaythorpe et al. to lie within a product-space estimation framework (*Gaythorpe et al., 2019*). All estimation was performed in R and convergence of the chains was checked visually. To produce the burden estimates, 200 samples of the posterior predictive distributions of the FOI in each province were taken which we then use to calculate the incidence of infections in each province.

We use published values of the proportion of infections which are severe and of the CFR to calculate the burden of disease. These proportions, estimated by *Johansson et al., 2014*, are that 12% [5%, 26%] of infections are severe and that 47% [31%, 62%] of severe infections result in death. However, these estimates remain uncertain since the disease is notoriously misdiagnosed and under-reported. Another area resulting in uncertainty in burden estimates is the heterogeneity in data availability, specifically serological surveys which are not currently available in West Africa or South America. As such, whilst the updated data and model averaging framework have improved the uncertainty ranges, these are still broad.

## Yellow Fever – University of Notre Dame

The University of Notre Dame yellow fever (YFV) model is a static transmission model that assumes a constant force of infection (FOI) for each endemic country. Yellow fever infections in the human population are thus modeled as spillover events from non-human primates, so human-to-human transmission observed in urban outbreaks is not considered. Accordingly, our model is intended to capture long-term changes in YFV burden on account of changes in vaccination coverage rather than to realistically capture interannual variability due to YFV epizootics in non-human primates and occasional outbreaks in humans.

We calibrated our YFV transmission model to multiple sources of epidemiological data collected in sub-Saharan Africa at the first administrative level sub-nationally. First, we quantified past exposure to YFV by estimating the force of infection in 23 administrative units using data collected in serological surveys. We then related the predicted number of YFV infections at each of the 23 administrative units to the corresponding reported outbreak data collated by *Garske et al., 2014* to quantify the extent of underreporting. We then obtained estimates of the total number of infections at each administrative unit in sub-Saharan Africa by relating our estimates of underreporting to the total number of reported cases and deaths in each administrative unit. This allowed us to estimate a posterior distribution of a single FOI for each administrative unit in sub-Saharan Africa. Because the

FOIs that we estimated are sensitive to the number of reported cases and deaths, we smoothed across our estimates by performing a regression analysis with spatial covariates. We considered multiple regression models and generated an ensemble prediction by weighting the predicted FOI from each regression model based on performance in ten-fold cross-validation at the country level. National-level FOI estimates were obtained by weighting the ensemble spatial prediction of FOI according to WorldPop 2015 population density estimates at the first administrative level and then summing to obtain national FOIs.

To project the number of yellow fever cases and deaths in each country under a given vaccination coverage scenario, we first scaled the national-level FOI by the proportion of the population that is unvaccinated. We then used the scaled FOI estimate to project the annual number of YFV infections and multiplied this quantity by the probabilities of disease and death reported by *Johansson et al., 2014* to obtain estimates of the annual number of YFV cases and deaths. We assume a 0.975 probability of protection from infection among those who are vaccinated based on *Jean et al., 2016*, with this level of protection assumed to be lifelong based on a single dose. In the event of campaigns, we assume that individuals are vaccinated randomly and irrespective of prior vaccination through another campaign or routine vaccination.

## Appendix 3

### 3.1 Impact by year of vaccination

Taking an activity perspective, we assume that RI and NRI, which target multiple age groups, have different effects; for example due to dosage clustering. Therefore, there are two activity-specific impact ratios which can then be multiplied by the number of FVPs to calculate impact provided by vaccination occurring in one year.

For RI, the impact ratio is defined as the impact for all cohorts who are vaccinated over time period $Y_v$ per the additional FVPs between the baseline and focal scenarios. The impact for RI, $D_R$, is given by $D_{b_R - f_R}$, where $b_R$ and $f_R$ are the baseline and focal RI scenarios, respectively. Here, the impact ratio for RI is given by the following:

$$\rho_R(c) = \frac{\sum_{y-a \in Y_v - A_v} D_R(a,c,y)}{\sum_{y \in Y_v} \mathrm{FVP}_R(c,y)}, \tag{C.1}$$

where $Y_v - A_v$ are cohorts receiving vaccinations in years $Y_v$. The impact by year of vaccination is then given by the following:

$$D_R(c,y) = \rho_R(c) \times \mathrm{FVP}_R(c,y), \tag{C.2}$$

where $FVP_R$ are FVPs vaccinated through RI.

For NRI, the impact ratio is averaged evenly over all ages across the entire time period ($Y_m$). This is because we do not attempt to predict future NRI coverage after the final year of credible campaign schedules. Therefore, the only impact due to NRI comes from NRI years $Y_v$ and all campaign impact for birth cohorts born after this period can be attributed back to these vaccination years. The impact of NRI, $D_S$, is given by $D_{b_S - f_S}$, where $b_S$ and $f_S$ are the baseline and focal NRI scenarios, respectively. The impact ratio is given by the following:

$$\rho_S(c) = \frac{\sum_{y \in Y_m} \sum_{a \in A_m} D_S(a,c,y)}{\sum_{y \in Y_v} \mathrm{FVP}_S(c,y)}. \tag{C.3}$$

The impact by year of vaccination is then given by the following:

$$D_S(c,y) = \rho_S(c) \times \mathrm{FVP}_S(c,y), \tag{C.4}$$

where $FVP_S$ are FVPs vaccinated through NRI.

The aggregated impact by YoV for both activities is the sum of the impact from RI and NRI, that is sum of Equations C.2 and C.4.

Further analysis on this method has been done (*Echeverria-Londono et al., 2021*).

## Appendix 4

### Specific differences to previous VIMC-wide study

#### 4.1 Fully vaccinated persons

Due to increases in coverage projections for years 2018 onward, the FVPs generally increased between the two model estimates (*Appendix 5—figure 3*). This is particularly seen for HPV where the optimistic assumptions around vaccine introductions leads to large numbers of FVPs, thereby an increase in deaths averted by cohort year, due to campaigns (*Appendix 5—figure 2*). As the models for HPV have remained fairly static, the difference in HPV vaccine impact estimates are due only to this change in FVPs.

#### 4.1 Model structure

The large changes in HepB, measles and YF estimates are due to changes in model structure since the *Li et al., 2019* study.

#### 4.2 Measles

The two measles models both reassessed the case fatality ratios (CFRs) used in their estimates. Previously, the CFRs were derived from *Wolfson et al., 2009*. However, a recent publication re-evaluated the case fatality ratio for measles and predicted how the CFR may change in future (*Portnoy et al., 2019*). They found that the CFR due to measles was likely lower than that of Wolfson et al. and expected to fall towards 2030. This means that deaths due to measles are expected to fall and thus, vaccine impact.

#### 4.3 Hepatitis B

The change in the estimates for HepB are driven by model structure alterations. These changes, detailed in the supplementary material for HepB, Imperial, take into account more optimistic assumptions around treatment in future years. As such, the burden and severe outcomes are reduced and the vaccine impact is also lowered.

#### 4.4 Yellow fever

In a change to the estimates shown in the previous VIMC-wide study, there are two models included in the current work for YF (*Li et al., 2019*). As such, the mean estimates will be affected by both models. The model described in *Appendix 5—figure 2* for YF, Imperial, was extensively expanded from the original work of *Garske et al., 2014* which included updated serological and outbreak data as well as new environmental covariates (*Gaythorpe et al., 2021a*). This lead to a decrease in overall uncertainty and thus a decrease in the mean expected burden and vaccine impact. This also affected the geographic distribution with the Democratic Republic of the Congo particularly highlighted for YF burden.

## Appendix 5

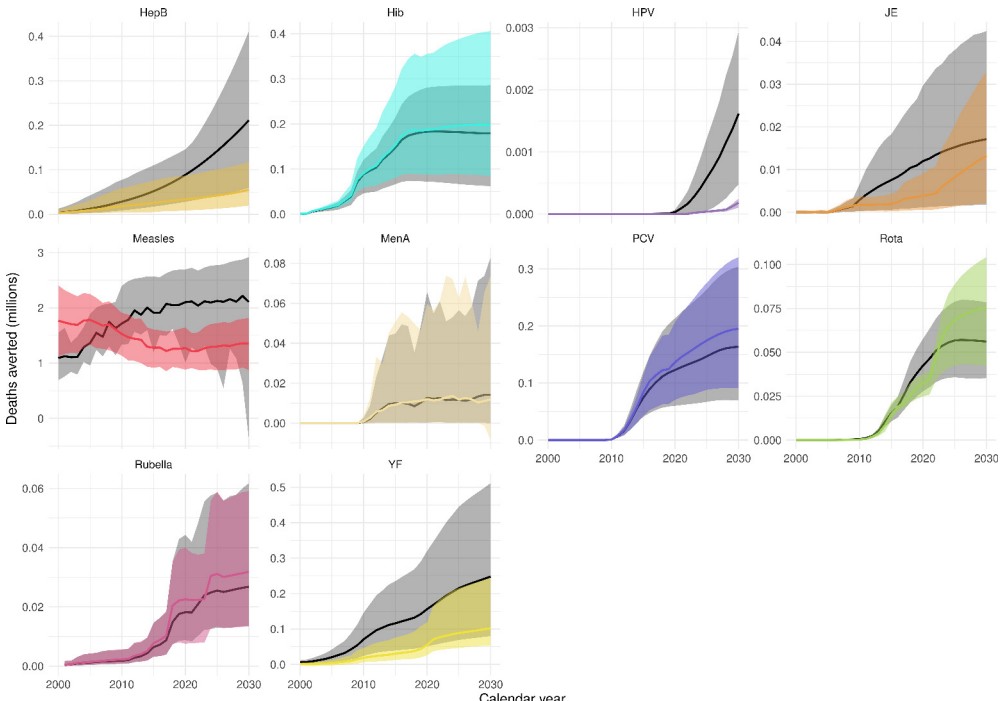

**Appendix 5—figure 1.** Deaths averted per calendar year for hepatitis B (HepB), *Haemophilus influenzae* type b (Hib), human papillomavirus (HPV), Japanese encephalitis (JE), measles, *Neisseria meningitidis* serogroup A (MenA), *Streptococcus pneumoniae* (PCV), rotavirus (Rota), rubella and yellow fever (YF). Coloured lines and areas indicate estimates based on this Vaccine Impact Modelling Consortium (VIMC) study and grey lines and areas indicate estimates based on previous VIMC results (*Li et al., 2019*). Ribbons indicate 95% CI.

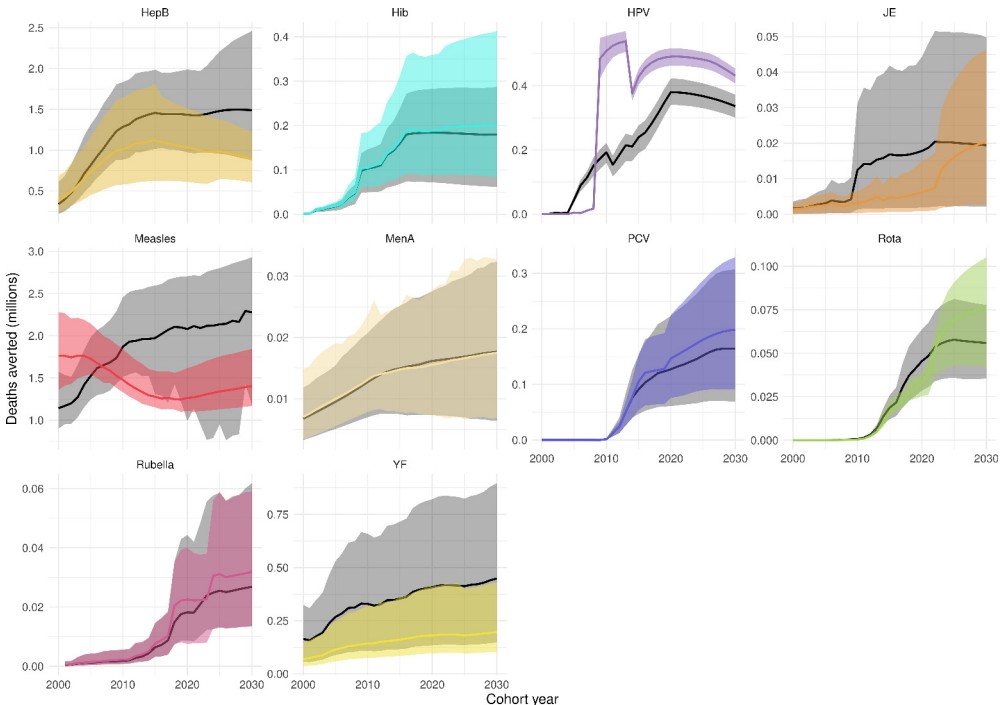

**Appendix 5—figure 2.** Deaths averted per birth cohort year for hepatitis B (HepB), *Haemophilus influenzae* type b (Hib), human papillomavirus (HPV), Japanese encephalitis (JE), measles, *Neisseria meningitidis* serogroup A (MenA), *Streptococcus pneumoniae* (PCV), rotavirus (Rota), rubella and yellow fever (YF). Coloured lines and areas indicate estimates based on this Vaccine Impact Modelling Consortium (VIMC) study and grey lines and areas indicate estimates based on previous VIMC results (*Li et al., 2019*). Ribbons indicate 95% CI.

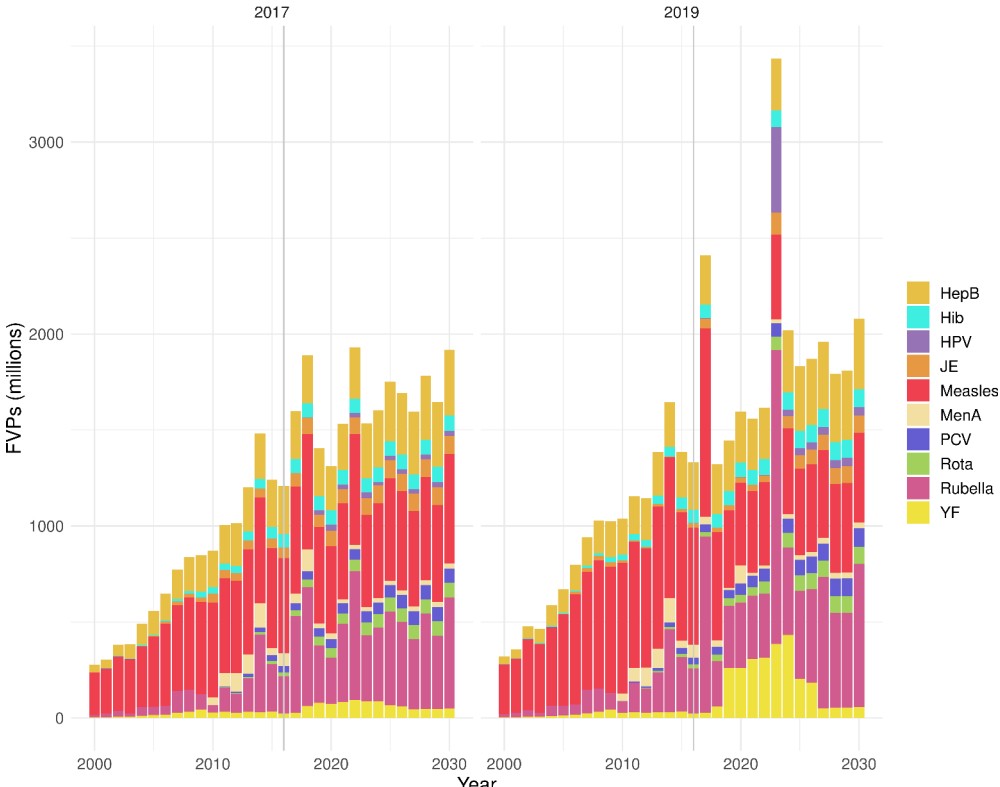

**Appendix 5—figure 3.** Comparison of fully vaccinated persons (FVPs) in millions between 2017 and 2019 model estimates used within the previous Vaccine Impact Modelling Consortium (VIMC)-wide study *Li et al., 2019* and this study, respectively. FVPs shown for hepatitis B (HepB), *Haemophilus influenzae* type b (Hib), human papillomavirus (HPV), Japanese encephalitis (JE), measles, *Neisseria meningitidis* serogroup A (MenA), *Streptococcus pneumoniae* (PCV), rotavirus (Rota), rubella, and yellow fever (YF).

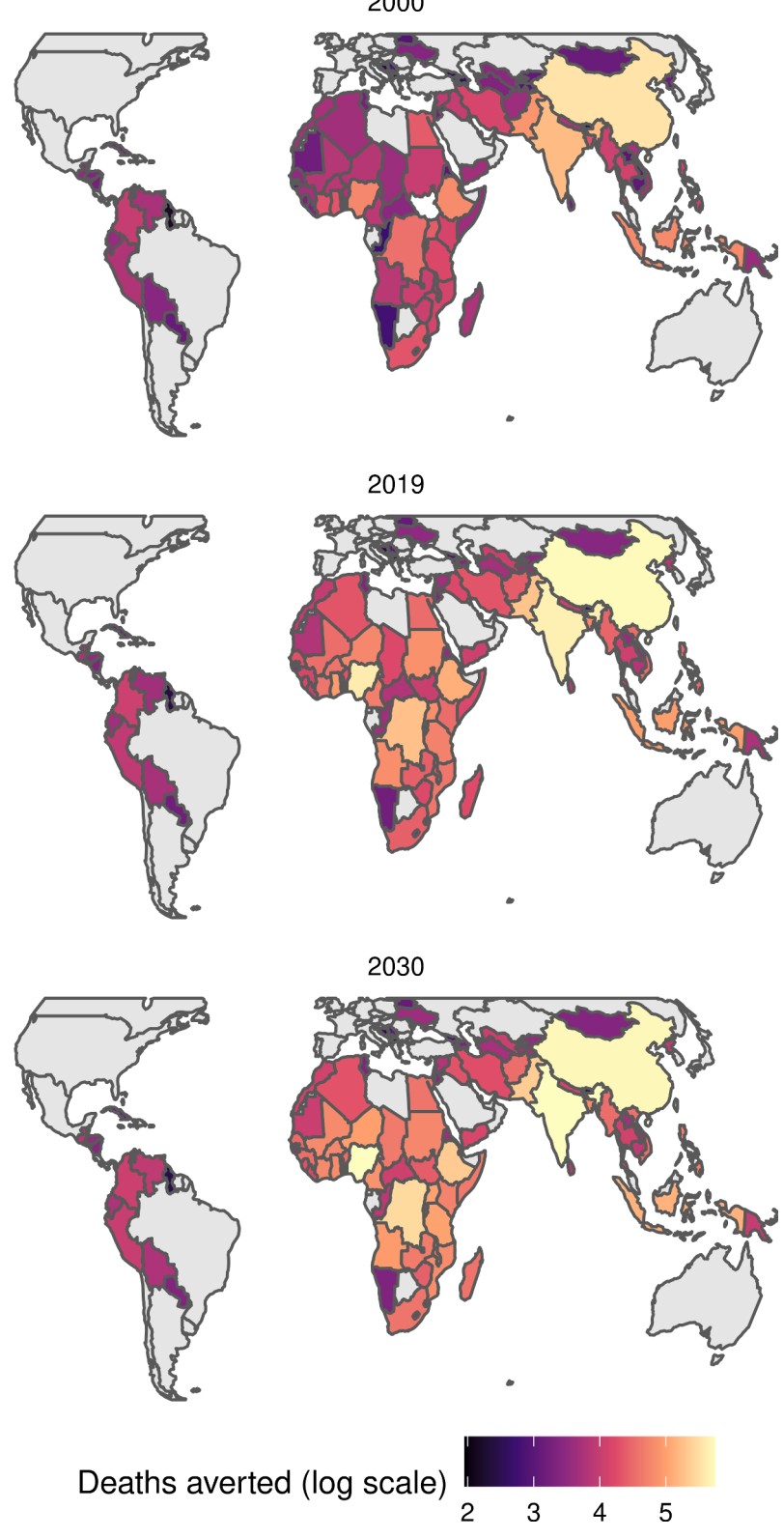

**Appendix 5—figure 4.** Estimated number of deaths averted per year of vaccination in 2000, 2019, and 2030 for all 10 Vaccine Impact Modelling Consortium (VIMC) pathogens.

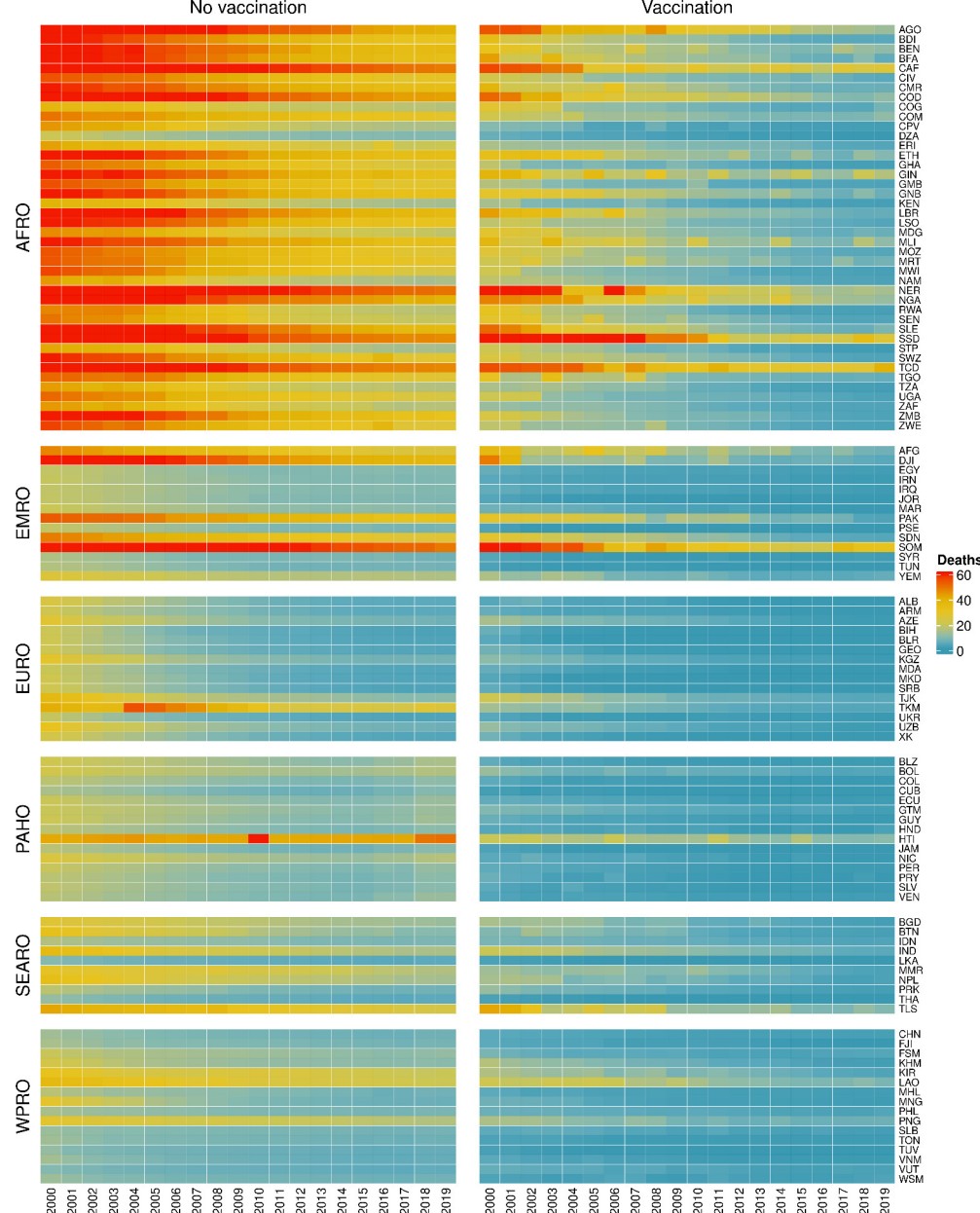

**Appendix 5—figure 5.** Mean predicted deaths for children under-5 with and without vaccination due to the 10 Vaccine Impact Modelling Consortium (VIMC) pathogens per 1000 lives per country for years 2000–2019. Countries are arranged by World Health Organisation (WHO) African (AFRO), Eastern Mediterranean (EMRO), European (EURO), Pan American (PAHO), South-East Asian (SEARO), and Western Pacific (WPRO) regions.

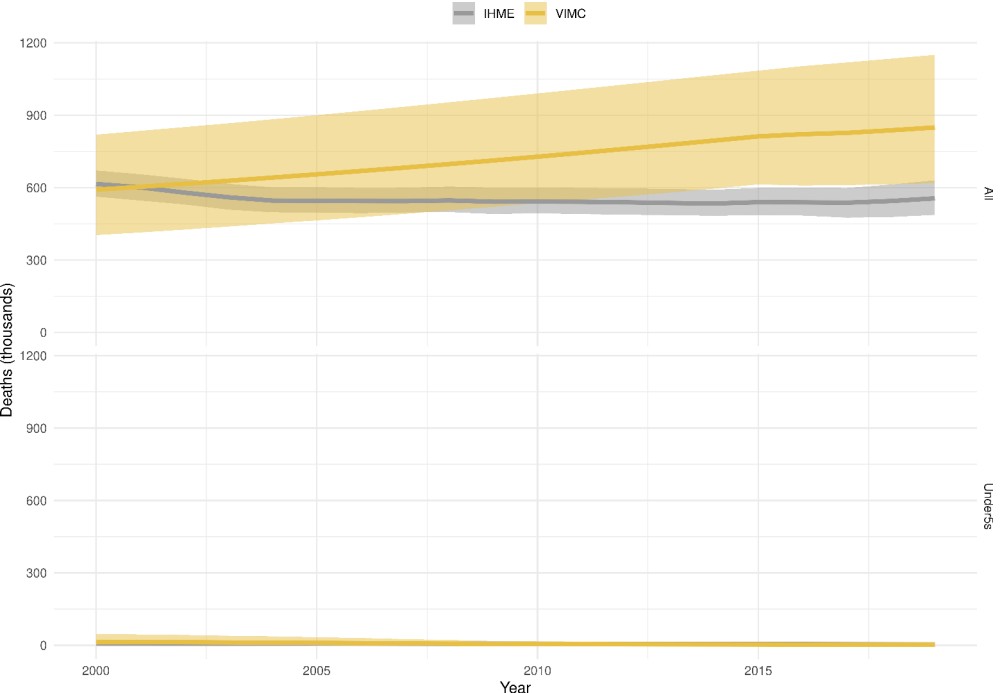

**Appendix 5—figure 6.** Global deaths for hepatitis B per calendar year (in thousands) for all ages and for children under-5. Orange lines and areas indicate estimates from the Vaccine Impact Modelling Consortium (VIMC). Grey lines and areas indicate estimates from the Global Burden of Disease Study (GBD) 2019 from the Institute for Health Metrics and Evaluation (IHME). Ribbons indicate 95% CI.

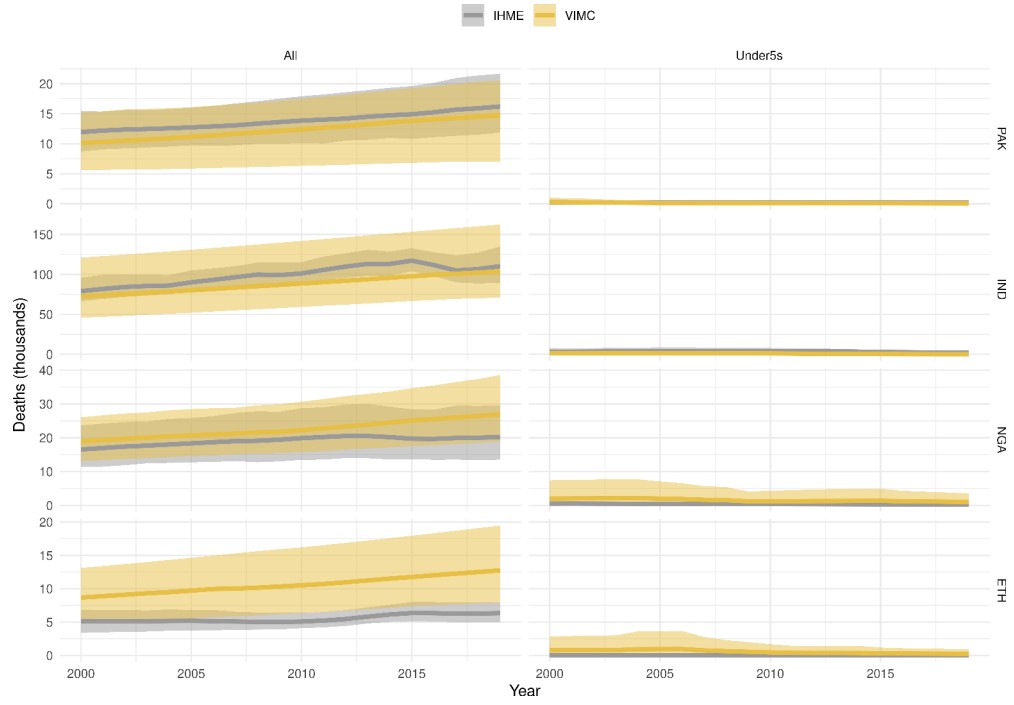

**Appendix 5—figure 7.** Deaths for hepatitis B in the PINE countries (Pakistan - PAK, India - IND,
*Appendix 5—figure 7 continued on next page*

*Appendix 5—figure 7 continued*

Nigeria - NGA and Ethiopia - ETH) per calendar year (in thousands) for all ages and for under-5s. Orange lines and areas indicate estimates from the Vaccine Impact Modelling Consortium (VIMC). Grey lines and areas indicate estimates from the Global Burden of Disease Study (GBD) 2019 from the Institute for Health Metrics and Evaluation (IHME). Ribbons indicate 95% CI.

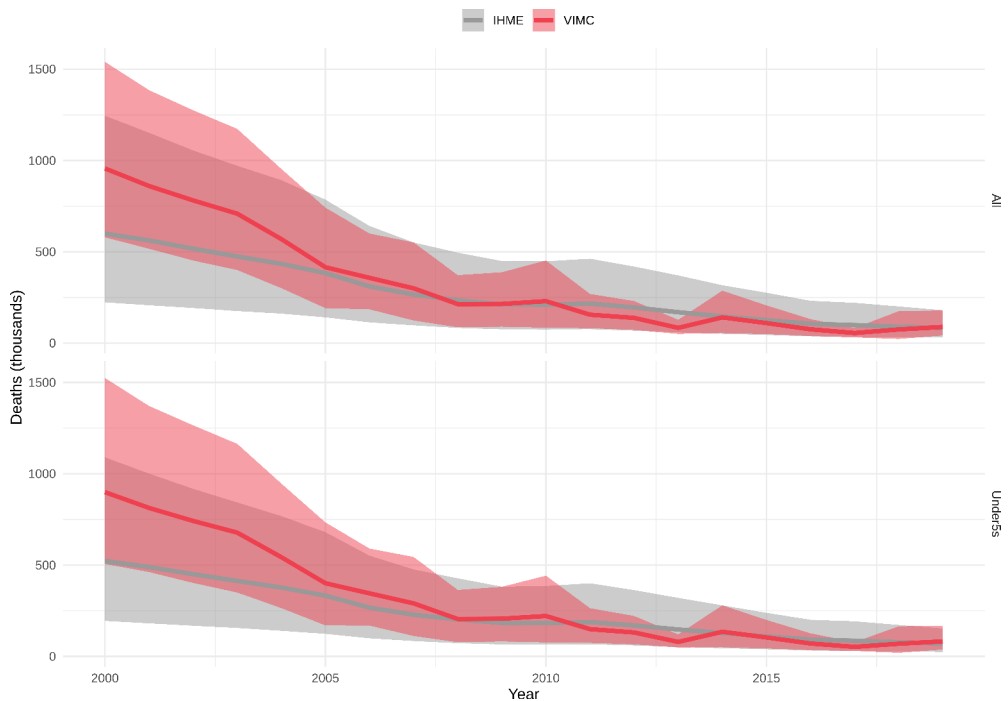

**Appendix 5—figure 8.** Global deaths for measles per calendar year (in thousands) for all ages and for children under-5. Red lines and areas indicate estimates from the Vaccine Impact Modelling Consortium (VIMC). Grey lines and areas indicate estimates from the Global Burden of Disease Study (GBD) 2019 from the Institute for Health Metrics and Evaluation (IHME). Ribbons indicate 95% CI.

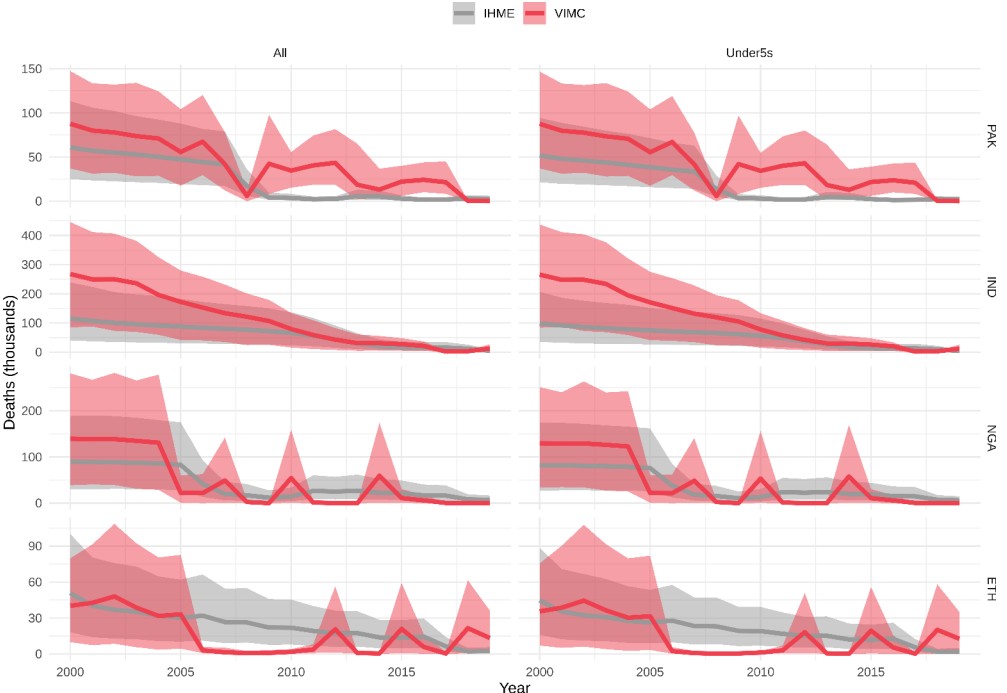

**Appendix 5—figure 9.** Deaths for measles in the PINE countries (Pakistan - PAK, India - IND, Nigeria - NGA and Ethiopia - ETH) per calendar year (in thousands) for all ages and for children under-5. Red lines and areas indicate estimates from the Vaccine Impact Modelling Consortium (VIMC). Grey lines and areas indicate estimates from the Global Burden of Disease Study (GBD) 2019 from the Institute for Health Metrics and Evaluation (IHME). Ribbons indicate 95% CI.

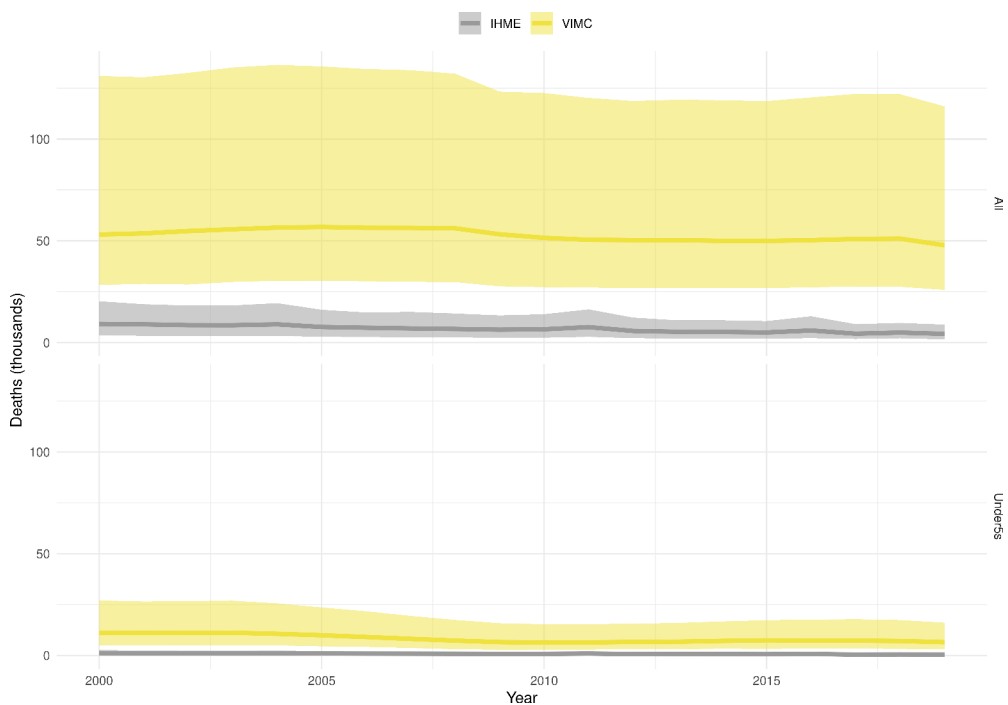

**Appendix 5—figure 10.** Global deaths for yellow fever per calendar year (in thousands) for all ages

*Appendix 5—figure 10 continued on next page*

*Appendix 5—figure 10 continued*

and for children under-5. Yellow lines and areas indicate estimates from the Vaccine Impact Modelling Consortium (VIMC). Grey lines and areas indicate estimates from the Global Burden of Disease Study (GBD) 2019 from the Institute for Health Metrics and Evaluation (IHME). Ribbons indicate 95% CI.

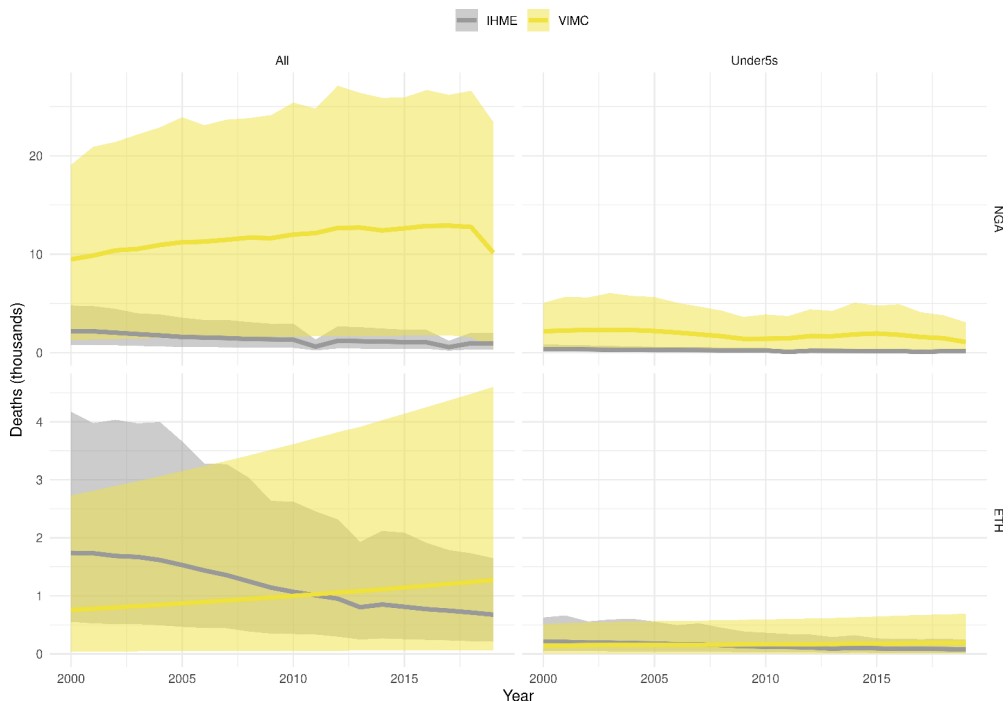

**Appendix 5—figure 11.** Deaths for yellow fever in Nigeria - NGA and Ethiopia - ETH per calendar year (in thousands) for all ages and for children under-5. Yellow lines and areas indicate estimates from the Vaccine Impact Modelling Consortium (VIMC). Grey lines and areas indicate estimates from the Global Burden of Disease Study (GBD) 2019 from the Institute for Health Metrics and Evaluation (IHME). Ribbons indicate 95% CI.

HepB

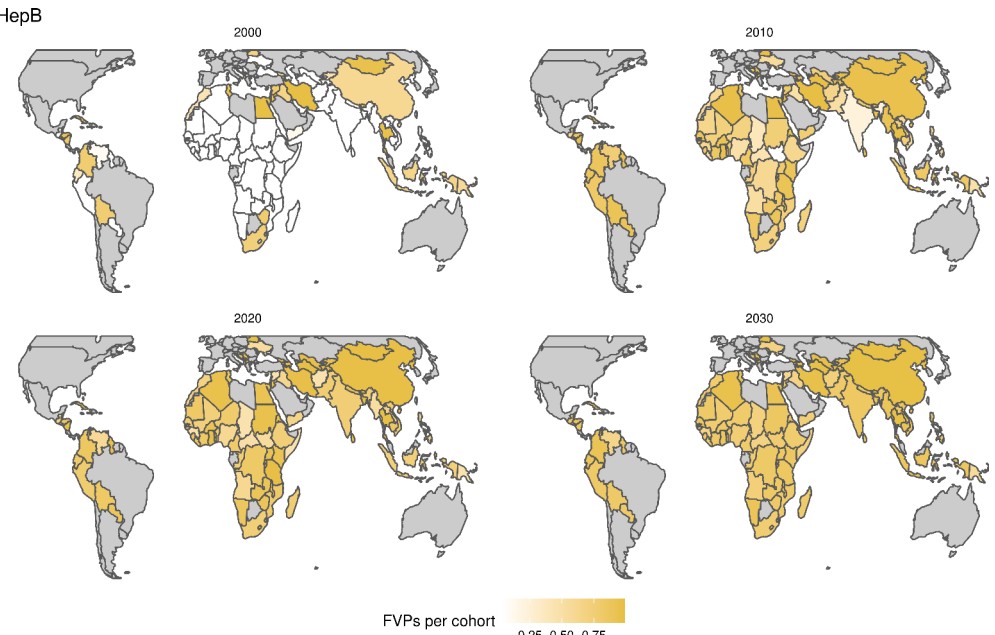

**Appendix 5—figure 12.** Fully vaccinated persons (FVPs) per vaccination target population for Hepatitis B for years 2000, 2010, 2020, and 2030.

Hib

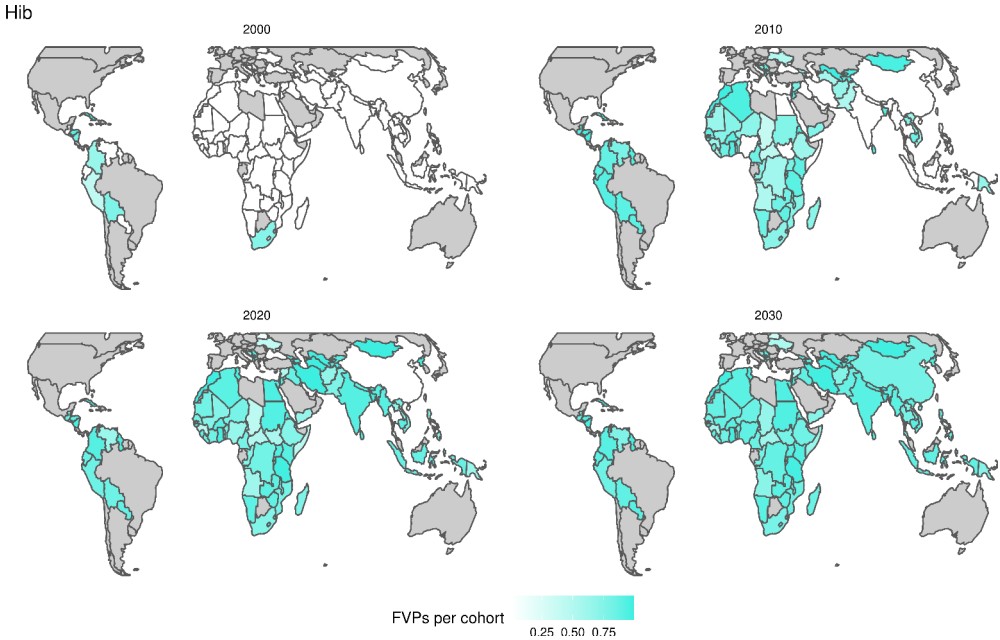

**Appendix 5—figure 13.** Fully vaccinated persons (FVPs) per vaccination target population for *Haemophilus influenzae* type b (Hib) for years 2000, 2010, 2020, and 2030.

Measles

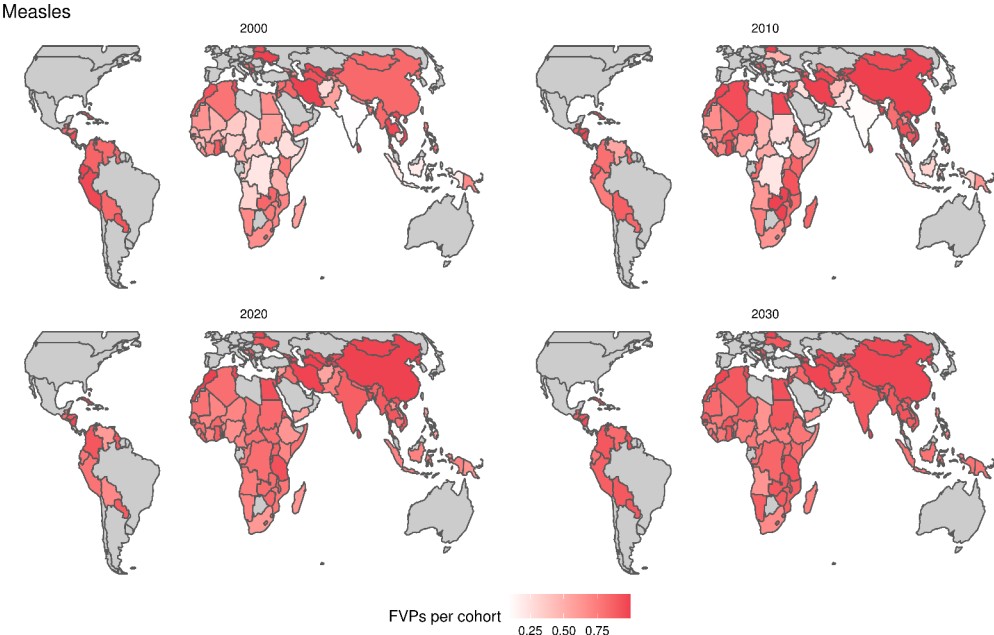

**Appendix 5—figure 14.** Fully vaccinated persons (FVPs) per vaccination target population for measles for years 2000, 2010, 2020, and 2030.

PCV

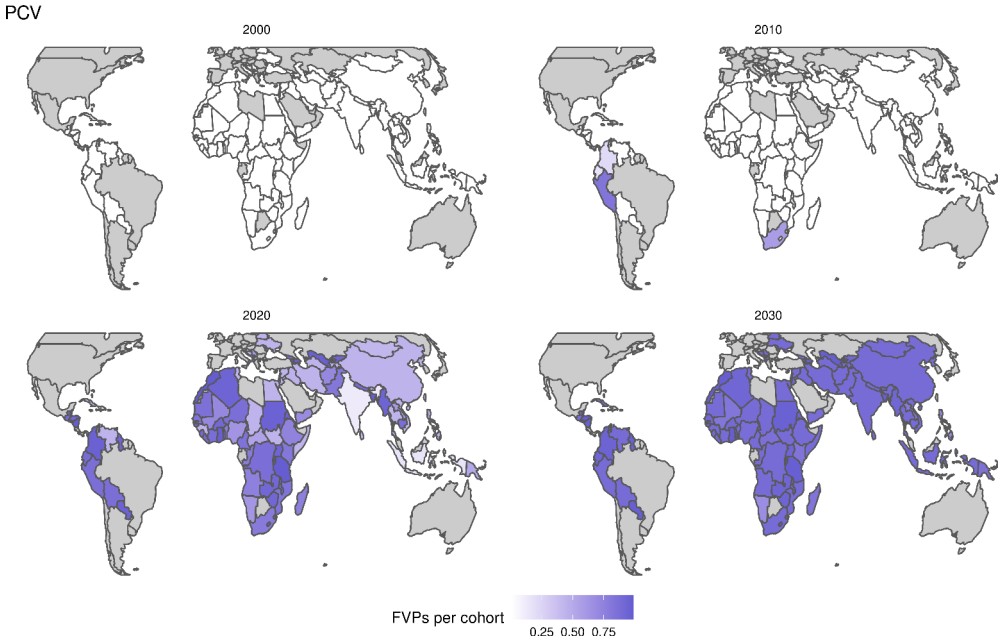

**Appendix 5—figure 15.** Fully vaccinated persons (FVPs) per vaccination target population for *Streptococcus pneumoniae* (PCV) for years 2000, 2010, 2020, and 2030.

Rota

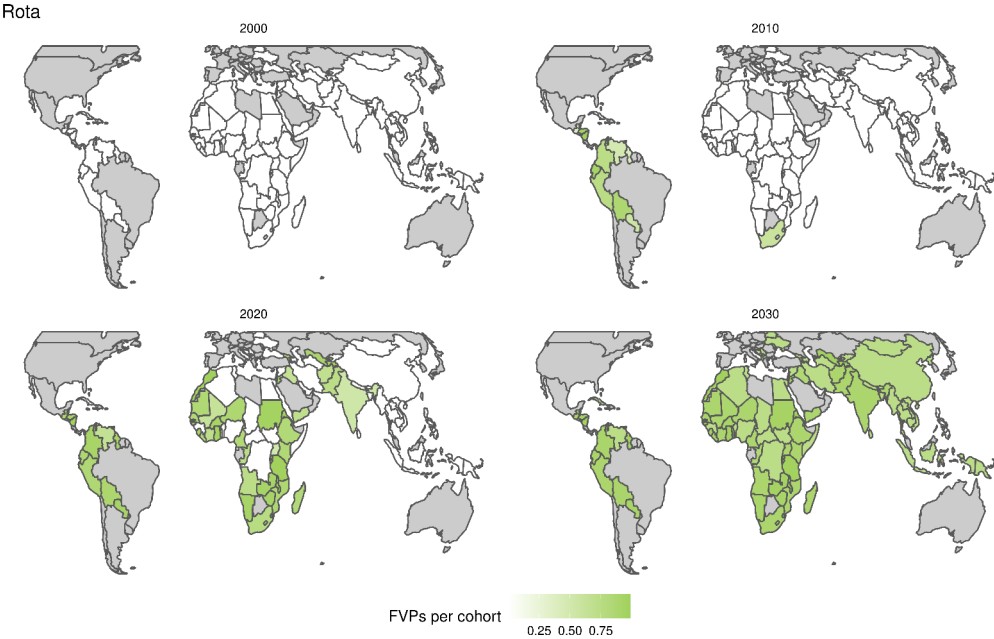

**Appendix 5—figure 16.** Fully vaccinated persons (FVPs) per vaccination target population for rotavirus for years 2000, 2010, 2020, and 2030.

HPV

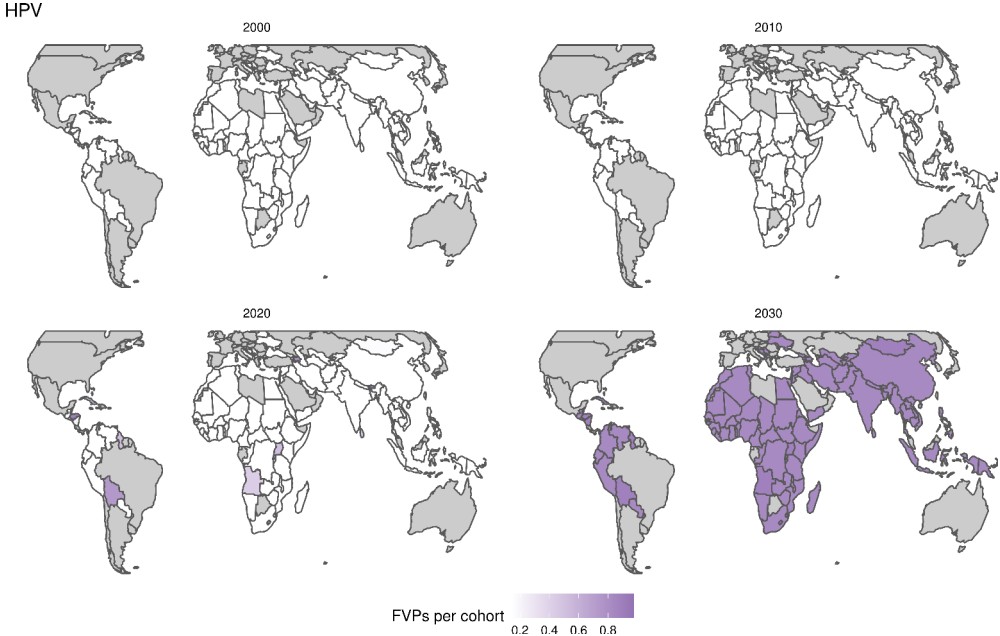

**Appendix 5—figure 17.** Fully vaccinated persons (FVPs) per vaccination target population for human papillomavirus (HPV) for years 2000, 2010, 2020, and 2030.

Rubella

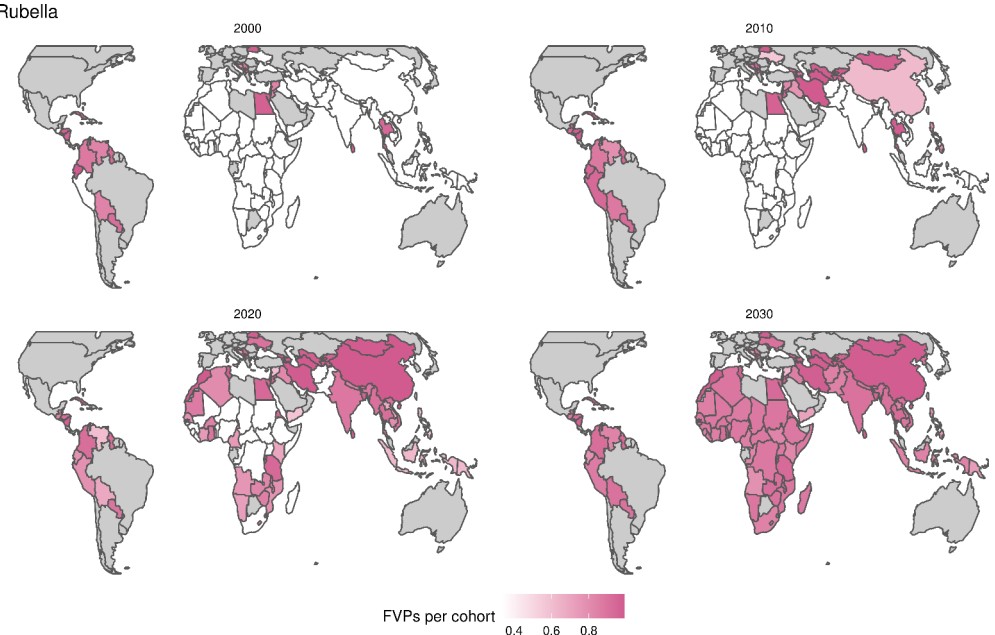

**Appendix 5—figure 18.** Fully vaccinated persons (FVPs) per vaccination target population for rubella for years 2000, 2010, 2020, and 2030.

YF

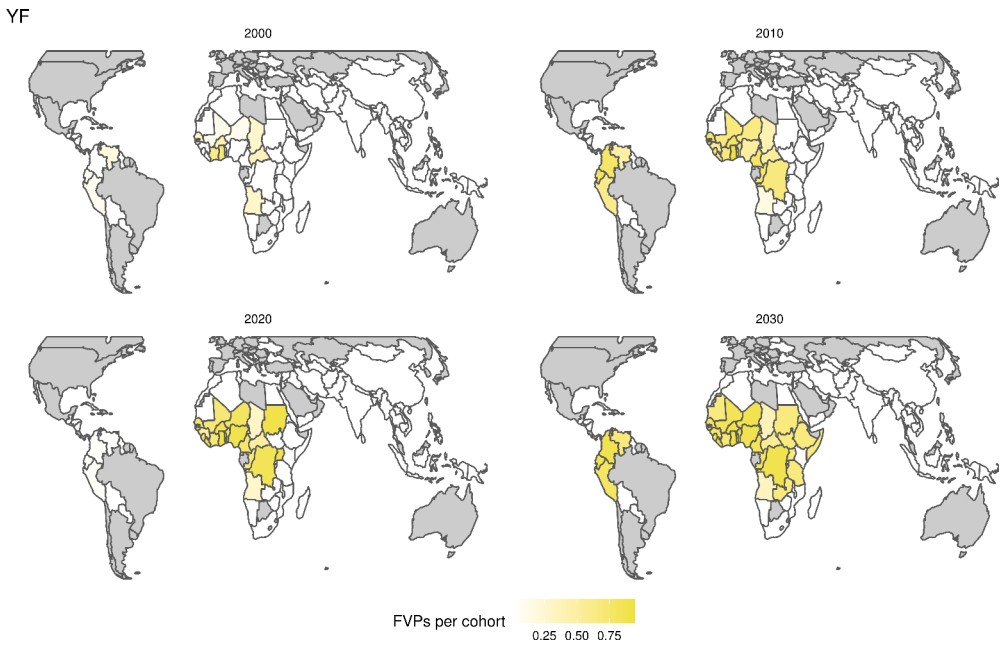

**Appendix 5—figure 19.** Fully vaccinated persons (FVPs) per vaccination target population for yellow fever (YF) for years 2000, 2010, 2020, and 2030.

MenA

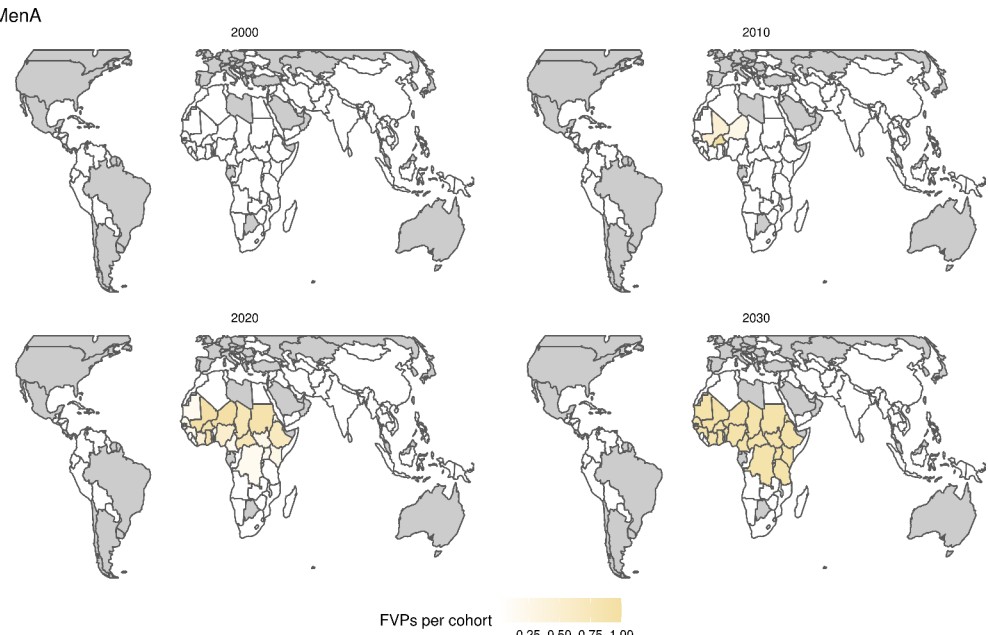

**Appendix 5—figure 20.** Fully vaccinated persons (FVPs) per vaccination target population for *Neisseria meningitidis* serogroup A (MenA) for years 2000, 2010, 2020, and 2030.

JE

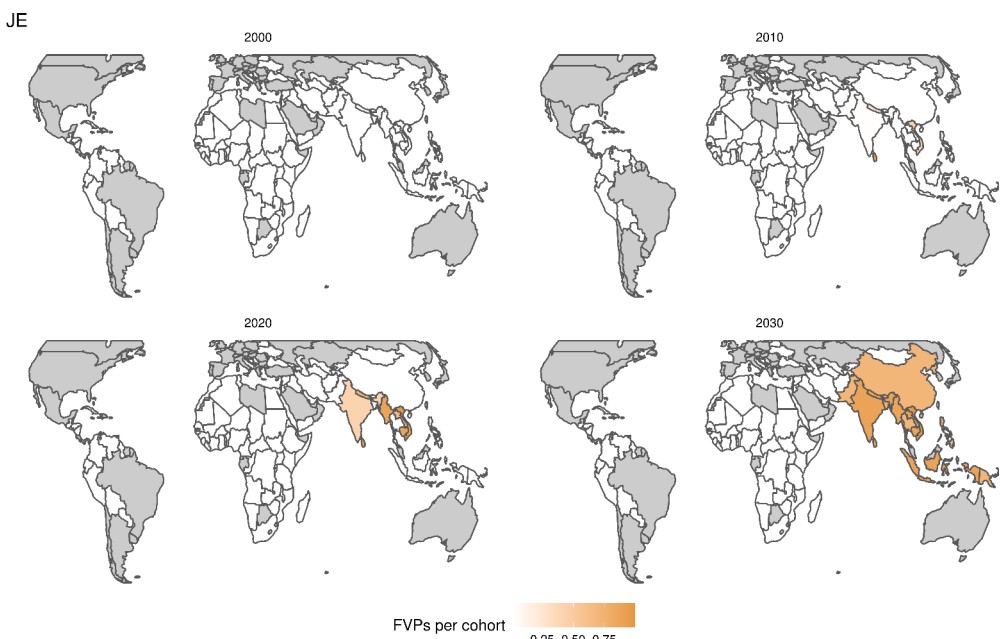

**Appendix 5—figure 21.** Fully vaccinated persons (FVPs) per vaccination target population for Japanese encephalitis (JE) for years 2000, 2010, 2020, and 2030.

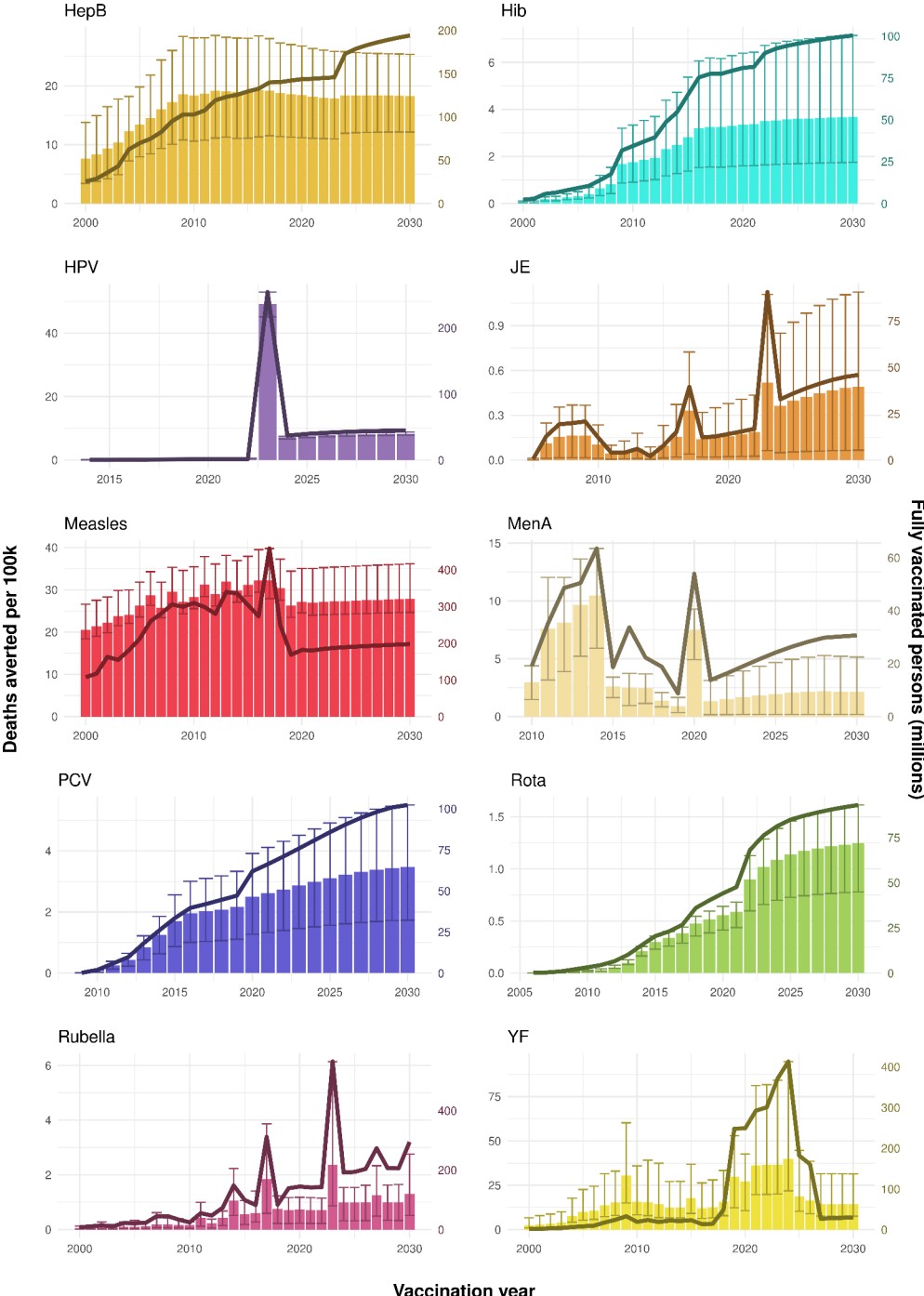

**Appendix 5—figure 22.** Deaths averted per 100,000 population per year of vaccination for hepatitis B (HepB), *Haemophilus influenzae* type b (Hib), human papillomavirus (HPV), Japanese encephalitis (JE), measles, *Neisseria meningitidis* serogroup A (MenA), *Streptococcus pneumoniae* (PCV), rotavirus (Rota), rubella, and yellow fever (YF). The bars show the number of deaths averted (per 100,000 population) in each vaccination year. Error bars indicate 95% CI. The line shows the number of fully vaccinated persons (FVPs; in millions) achieved in each year's vaccination activities.

## Appendix 6

The full list of countries included in this and the previous VIMC studies is provided in table *Appendix 6—table 1*

**Appendix 6—table 1.** 112 countries included in the analysis.

Those TRUE for gavi73 receive GAVI support; those TRUE for vimc98 were included in the previous VIMC-wide study (*Li et al., 2019*).

| Country | Country name | gavi73 | vimc98 |
|---------|--------------|--------|--------|
| AFG | Afghanistan | TRUE | TRUE |
| AGO | Angola | TRUE | TRUE |
| ALB | Albania | FALSE | TRUE |
| ARM | Armenia | TRUE | TRUE |
| AZE | Azerbaijan | TRUE | TRUE |
| BDI | Burundi | TRUE | TRUE |
| BEN | Benin | TRUE | TRUE |
| BFA | Burkina Faso | TRUE | TRUE |
| BGD | Bangladesh | TRUE | TRUE |
| BIH | Bosnia and Herzegovina | FALSE | TRUE |
| BLR | Belarus | FALSE | FALSE |
| BLZ | Belize | FALSE | TRUE |
| BOL | Bolivia, Plurinational State of | TRUE | TRUE |
| BTN | Bhutan | TRUE | TRUE |
| CAF | Central African Republic | TRUE | TRUE |
| CHN | China | FALSE | TRUE |
| CIV | Cote d'Ivoire | TRUE | TRUE |
| CMR | Cameroon | TRUE | TRUE |
| COD | Congo, the Democratic Republic of the | TRUE | TRUE |
| COG | Congo | TRUE | TRUE |
| COL | Colombia | FALSE | FALSE |
| COM | Comoros | TRUE | TRUE |
| CPV | Cabo Verde | FALSE | TRUE |
| CUB | Cuba | TRUE | TRUE |
| DJI | Djibouti | TRUE | TRUE |
| DZA | Algeria | FALSE | FALSE |
| ECU | Ecuador | FALSE | FALSE |
| EGY | Egypt | FALSE | TRUE |
| ERI | Eritrea | TRUE | TRUE |
| ETH | Ethiopia | TRUE | TRUE |
| FJI | Fiji | FALSE | TRUE |
| FSM | Micronesia, Federated States of | FALSE | TRUE |
| GEO | Georgia | TRUE | TRUE |
| GHA | Ghana | TRUE | TRUE |
| GIN | Guinea | TRUE | TRUE |
| GMB | Gambia | TRUE | TRUE |
| GNB | Guinea-Bissau | TRUE | TRUE |
| GTM | Guatemala | FALSE | TRUE |

*Continued on next page*

*Appendix 6—table 1 continued*

| Country | Country name | gavi73 | vimc98 |
| --- | --- | --- | --- |
| GUY | Guyana | TRUE | TRUE |
| HND | Honduras | TRUE | TRUE |
| HTI | Haiti | TRUE | TRUE |
| IDN | Indonesia | TRUE | TRUE |
| IND | India | TRUE | TRUE |
| IRN | Iran, Islamic Republic of | FALSE | FALSE |
| IRQ | Iraq | FALSE | TRUE |
| JAM | Jamaica | FALSE | FALSE |
| JOR | Jordan | FALSE | FALSE |
| KEN | Kenya | TRUE | TRUE |
| KGZ | Kyrgyzstan | TRUE | TRUE |
| KHM | Cambodia | TRUE | TRUE |
| KIR | Kiribati | TRUE | TRUE |
| LAO | Lao People's Democratic Republic | TRUE | TRUE |
| LBR | Liberia | TRUE | TRUE |
| LKA | Sri Lanka | TRUE | TRUE |
| LSO | Lesotho | TRUE | TRUE |
| MAR | Morocco | FALSE | TRUE |
| MDA | Moldova, Republic of | TRUE | TRUE |
| MDG | Madagascar | TRUE | TRUE |
| MHL | Marshall Islands | FALSE | TRUE |
| MKD | Macedonia, the former Yugoslav Republic of | FALSE | FALSE |
| MLI | Mali | TRUE | TRUE |
| MMR | Myanmar | TRUE | TRUE |
| MNG | Mongolia | TRUE | TRUE |
| MOZ | Mozambique | TRUE | TRUE |
| MRT | Mauritania | TRUE | TRUE |
| MWI | Malawi | TRUE | TRUE |
| NAM | Namibia | FALSE | FALSE |
| NER | Niger | TRUE | TRUE |
| NGA | Nigeria | TRUE | TRUE |
| NIC | Nicaragua | TRUE | TRUE |
| NPL | Nepal | TRUE | TRUE |
| PAK | Pakistan | TRUE | TRUE |
| PER | Peru | FALSE | FALSE |
| PHL | Philippines | FALSE | TRUE |
| PNG | Papua New Guinea | TRUE | TRUE |
| PRK | Korea, Democratic People's Republic of | TRUE | TRUE |
| PRY | Paraguay | FALSE | TRUE |
| PSE | Palestine, State of | FALSE | TRUE |
| RWA | Rwanda | TRUE | TRUE |
| SDN | Sudan | TRUE | TRUE |

*Continued on next page*

*Appendix 6—table 1 continued*

| Country | Country name | gavi73 | vimc98 |
|---------|--------------|--------|--------|
| SEN | Senegal | TRUE | TRUE |
| SLB | Solomon Islands | TRUE | TRUE |
| SLE | Sierra Leone | TRUE | TRUE |
| SLV | El Salvador | FALSE | TRUE |
| SOM | Somalia | TRUE | TRUE |
| SRB | Serbia | FALSE | FALSE |
| SSD | South Sudan | TRUE | TRUE |
| STP | Sao Tome and Principe | TRUE | TRUE |
| SWZ | Swaziland | FALSE | TRUE |
| SYR | Syrian Arab Republic | FALSE | TRUE |
| TCD | Chad | TRUE | TRUE |
| TGO | Togo | TRUE | TRUE |
| THA | Thailand | FALSE | FALSE |
| TJK | Tajikistan | TRUE | TRUE |
| TKM | Turkmenistan | FALSE | TRUE |
| TLS | Timor-Leste | TRUE | TRUE |
| TON | Tonga | FALSE | TRUE |
| TUN | Tunisia | FALSE | TRUE |
| TUV | Tuvalu | FALSE | TRUE |
| TZA | Tanzania, United Republic of | TRUE | TRUE |
| UGA | Uganda | TRUE | TRUE |
| UKR | Ukraine | TRUE | TRUE |
| UZB | Uzbekistan | TRUE | TRUE |
| VEN | Venezuela, Bolivarian Republic of | FALSE | FALSE |
| VNM | Viet Nam | TRUE | TRUE |
| VUT | Vanuatu | FALSE | TRUE |
| WSM | Samoa | FALSE | TRUE |
| XK | Kosovo | FALSE | TRUE |
| YEM | Yemen | TRUE | TRUE |
| ZAF | South Africa | FALSE | FALSE |
| ZMB | Zambia | TRUE | TRUE |
| ZWE | Zimbabwe | TRUE | TRUE |

