## [Decision Letter]

**Acceptance summary:**

This is an important study. The methodology is robust, the data is robust. The results are of central importance in establishing the impact of vaccination prior to the Covid 19 pandemic with robust estimates of deaths averted or in vaccinated individuals in the time periods 2000-2019 or deaths which would be averted 2020-2030 and 2000-2030, emphasizing the importance of the achievements of vaccination activities to date and the need to sustain existing vaccination programmes.

**Decision letter after peer review:**

Congratulations, we are pleased to inform you that your article, "Lives saved with vaccination for 10 pathogens across 112 countries in a pre-COVID-19 world", has been accepted for publication in *eLife*. Your article has been reviewed by 3 peer reviewers, one of whom is a member of our Board of Reviewing Editors, and the evaluation has been overseen by a Senior Editor. The reviewers have opted to remain anonymous.

*Reviewer #1:*

Toor, Echeverria-Londono, Li et al., have delivered an ambitious and excellent collation and summary of complex mathematical models in order to estimate the impact of vaccination for 10 pathogens across 112 countries, representing the vast majority of the impact on vaccine preventable infectious disease worldwide.

Besides being a product of a comprehensive group of expert contributors, the application of a new method to quantify the impact of vaccination (lifetime impact attributable to a single year of activity) is a strength, as this gives a motivating assessment of impact.

The manuscript is well written throughout. It is also comprehensive and uses appendixes well to share more detailed content. One exception to this is a lack of content about the epidemiological assumptions of the models. Even if these are in Appendix B2, some description would be helpful in the text.

The authors address comparison with other work with similar ambitions very well.

The following two suggestions for additions to the discussion could improve the paper, if the authors and editors agree:

– The presentation of the findings as numbers of lives saved is impressive but is not put into context. Could the communication be strengthened by putting the estimates of lives saved into some context, either as a percentage of all deaths, or of lives lost due to these pathogens, or compared to lives saved by other healthcare interventions, for all ages and/or for under 5s? 98 million lives sounds like a great deal – but what percentage of infectious disease mortality (or just due to these pathogens) is this?

– Following on from the point above, could there be some mention in discussion of the potential for saving more lives in the 2020-2030 period through the introduction of other vaccines (COVID and other)? Unquantified is fine. It would be of interest to readers to know if there are any other vaccines approaching implementation that will likely reduce mortality due to infectious diseases even further during this time period. Possibly this is an extension to the project that is in progress? If so, that could be said?

– I could not find/access appendix B2 (apologies if my technical failing) – is this where the epi details are presented? Even if so, I think these need more prominence in the paper, as a key part of the basis for the impact of the vaccination activity that is described. And some discussion of the limitations in these parameters – across the models – as I imagine data for epidemiological parameters from some counties was generalised to others? And or was more reliable for some counties/models.

*Reviewer #2:*

This study reports data from the Vaccine impact modelling consortium (VIMC) on disease burden and estimate of vaccine impact in 112 countries against 10 pathogens hepatitis B (HepB), Haemophilus influenzae type b (Hib), human papillomavirus (HPV), Japanese encephalitis (JE), measles, Neisseria meningitides serogroup A (MenA), Streptococcus pneumoniae (PCV), rotavirus (Rota), rubella and yellow fever (YF). The data presented is an attempt by the VIMC to estimate the impact (number of lives saved) of a specific year's vaccination activities followed over the lifetime of those vaccinated. The strengths of the study are the use of 21 mathematical models -2 models per pathogen. The models vary in type and complexity giving a robustness to the data but also reflects the uncertainties inherent in the modelling of disease risks.

This is an important study. The methodology is robust, the data is robust. The results are of central importance in establishing the impact of vaccination prior to the Covid 19 pandemic with robust estimates of deaths averted or in vaccinated individuals in the time periods 2000-2019 or deaths which would be averted 2020-2030 and 2000-2030, emphasizing the importance of the achievements of vaccination activities to date and the need to sustain existing vaccination programmes.

The manuscript is lucid and well written. This is a complex subject to communicate but the authors succeed.

This is a clear exposition of a complex but crucial topic. The authors have critically assessed the data and the methodologies and the limitations of methods and the uncertainties of modelling are addressed.